# CHESSARENA: A CHESS TESTBED FOR EVALUATING STRATEGIC REASONING CAPABILITIES OF LARGE LANGUAGE MODELS

## ABSTRACT

Recent large language models (LLMs) have shown strong reasoning capabilities. However, a critical question remains: do these models possess genuine reasoning skills—particularly complex strategic reasoning—or are they primarily excelling at sophisticated pattern recognition within their training data? To address this question, this paper presents a chess testbed, ChessArena, to evaluate the strategic reasoning capabilities of LLMs. Chess requires complex strategic reasoning capabilities including long-term planning, strict rule comprehension, and multi-turn conversation memorization. Specifically, ChessArena is a competitive framework where LLMs play against each other, under four different play modes. The testbed is equipped with a ranking algorithm and a leaderboard. The testbed can also evaluate fine-grained capabilities including basic understanding, move selection, and puzzle solving. Over 13 LLMs with different modes are evaluated in ChessArena, playing over 800 games. The results reveal significant shortcomings in current LLMs: no model can beat Maia-1100 (a chess engine at human amateur level), while some even failed to defeat a random player that selects moves arbitrarily. We also present a strong baseline to the testbed: our fine-tuned Qwen3-8B substantially improved performance, approaching much larger state-of-the-art reasoning models.

## 1 INTRODUCTION

Large language models (LLMs) have demonstrated remarkable capabilities across diverse domains, from code generation (Jimenez et al., 2023) to mathematical problem-solving (Cobbe et al., 2021). One significant contributing factor to the success is the availability of high-quality benchmarks such as LiveCodeBench (Jain et al., 2024) and AIME2025 (MAA, 2025).

As LLMs are increasingly applied in real-world problems, improving their strategic reasoning capability, i.e., the reasoning under dynamic environments and uncertain adversary actions (Gandhi et al., 2023; Duan et al., 2024), becomes an urgent demand. However, there is still a lack of well-established evaluation frameworks for effectively evaluating the strategic reasoning capabilities of LLMs. Existing evaluation frameworks typically focus on isolated question-answering tasks that may not capture the essential aspects of strategic reasoning (Lin et al., 2025; Kazemi et al., 2025; Dua et al., 2019; Chen et al., 2021). Additionally, current benchmarks (Austin et al., 2021; Sprague et al.) often suffer from data contamination, where test examples may have appeared in training data.

In this work, we choose chess as the testbed for evaluating the strategic reasoning capability of LLMs, as it provides an ideal environment requiring the ability to maintain coherent strategies across prolonged gameplay, follow complex instructions consistently, and adapt reasoning based on evolving contexts. Additionally, the vast state space of chess–with an estimated $10^{47}$ possible board positions–virtually eliminates data contamination concerns.

Building on these advantages, we introduce ChessArena, a competitive platform where LLMs engage in complete chess games from opening to endgame. Our system implements a comprehensive ranking mechanism using approximately 30 games per model to ensure stable performance assessment. We evaluate models across four distinct play modes–Bullet, Blitz, Standard, and Blindfold–each designed to test different aspects of model capability, from rapid decision-making to memory retention in long-term contexts.

Our evaluation of over 13 state-of-the-art models, including O3, Gemini-2.5-Pro, and Doubao-Seed-1-6-Thinking, reveals their significant limitations. No model successfully defeated Maia-1100, a chess engine designed to play at a human amateur level, with some models losing even to a random player that arbitrarily selects a move from all legal moves. These results highlight three critical deficiencies: inconsistent instruction following (failure to maintain proper output formatting), weak tactic reasoning (selecting moves inferior to random choices), and limited multi-turn coherence (inability to maintain consistent play across multi-round games).

To investigate the underlying causes of these performance gaps, we developed three complementary evaluation tasks targeting specific reasoning components: basic rule understanding, single-move evaluation, and multi-step puzzle solving. These fine-grained assessments, combined with the competitive arena, provide comprehensive insight into model capabilities and limitations.

Finally, we demonstrate that post-training can address some of these deficiencies. Using high-quality gameplay data collected from ChessArena competitions, we fine-tuned Qwen3-8B through supervised learning followed by reinforcement learning. The resulting Qwen3-8B-Chess model shows substantial improvements in chess performance.

Our work makes three primary contributions:

- **ChessArena Platform**: We introduce a competitive evaluation framework for chess play. It is extensible, providing interfaces to any LLM participants. It supports the evaluation of complete games as well as fine-grained tasks targeted at specific reasoning components.

- **Empirical Findings**: Our testbed exposes critical gaps in current LLMs' strategic reasoning through over 800 systematic gameplays. Fine-grained evaluations also provide detailed insight into the sources of model limitations.

- **Training Data and Model**: We collect and curate high-quality strategic reasoning data from ChessArena, and demonstrate through the Qwen3-8B-Chess model that training on strategic reasoning data can improve performance.

## 2 CHESSARENA

### 2.1 OVERVIEW

As shown in Fig. 1, ChessArena is a simulation platform where LLMs compete against each other to acquire quantitative chess strength ratings. Each model operates independently, generating moves based solely on the current chessboard state, closely emulating human competitive play. During gameplay, models receive task instructions and board representations, analyze the position, and predict moves that iteratively update the chessboard state. Additionally, our ChessArena competitions offer high scalability, with unified interface management that facilitates easy integration of new LLMs without affecting existing rankings. We use Forsyth-Edwards Notation (FEN) (rec.games.chess, 1994) for board representation and support both Universal Chess Interface (UCI) (Kahlen, 2004) and Standard Algebraic Notation (SAN) (rec.games.chess, 1994) for move representations. For more information about these representations, please refer to Appendix B.4.

### 2.2 PLAY MODES

To better evaluate the ability of LLMs, we design four play modes inspired by Lichess[1]. Each LLM player can be associated with one of the following four modes.

- **Bullet**: Given the chessboard state, the LLM must directly generate a move without any intermediate reasoning. Outputs containing any form of thinking process will be rejected.

- **Blitz**: Given the chessboard state, the LLM may optionally include a reasoning process before producing the move. This mode is designed specifically for non-thinking LLMs.

- **Standard**: Given the chessboard state, the LLM must generate a move accompanied by a chain-of-thought (CoT) reasoning process. This mode is designed specifically for thinking LLMs.

---

[1]https://lichess.org/

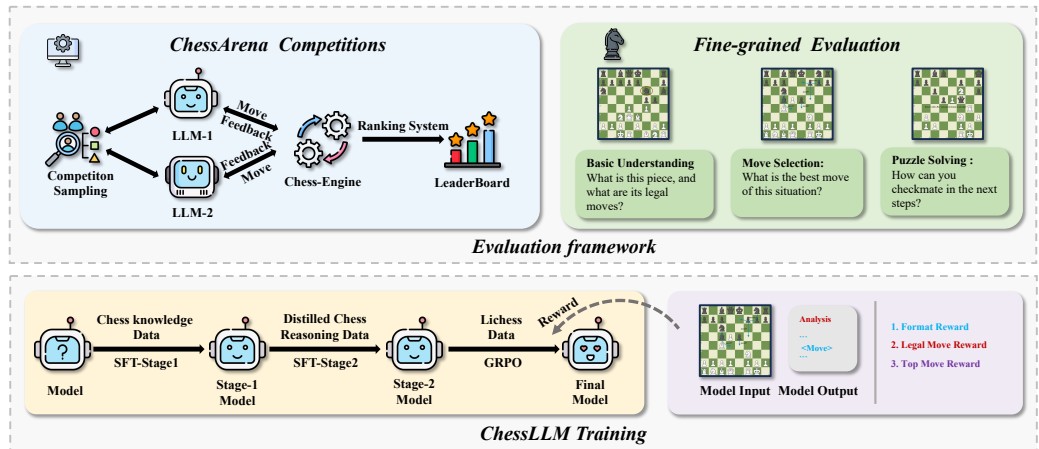

Figure 1: Overview of ChessArena competitions, fine-grained evaluation, and ChessLLM training. (1) An LLM can be integrated into ChessArena to compete against other models. After a certain number of competitions, each model is assigned a reliable Glicko rating and added to the leaderboard. (2) Three additional evaluation tasks are integrated into ChessArena to evaluate the chess capabilities at a fine-grained level. (3) We can extract high-quality chess reasoning data from the gameplay process, which can be used for training an LLM specially for chess.

- **Blindfold**: This mode represents the highest difficulty level. The model is provided with the move history from both players in the form of a multi-turn conversation. The LLM must reconstruct the chessboard state internally and produce a move with a thorough analysis.

## 2.3 RANKING SYSTEM

**Glicko Rating System**. We adopted the Glicko rating system (Glickman, 1995) as our ranking algorithm. As an enhancement of the traditional Elo rating system, Glicko represents each player's chess strength using two parameters: the rating $r$ (similar to traditional Elo) and the rating deviation $d$ that reflects the uncertainty in the rating. A high $d$ indicates that the player's rating is still unreliable and requires more matches to stabilize. In our scenarios, both parameters update after each competition, with $d$ decreasing monotonically as the system gains confidence in the player's skill level.

**Competition Sampling Strategy**. We developed a competition sampling algorithm to accelerate the convergence of rating deviation ($d$). Mathematical analysis shows that $d$ reduction is maximized when opponents have similar ratings ($r$) and low $d$ values, as matches between players of comparable and established skill levels yield the most information. Complete details of the Glicko rating system and proofs regarding the competition sampling strategy are provided in Appendix C. Our algorithm enables new players to achieve reliable ratings ($d < 100$) within approximately 30 games.

## 2.4 CHESS ENGINE

Regarding the chess engine, we chose Stockfish (Stockfish Development Team, 2016), which is currently the most powerful chess engine and has been widely used in chess analysis. We utilize the analysis results from Stockfish as a critical reference for subsequent evaluations. Specifically, given a search depth and a chessboard state, we use Stockfish to analyze the win rates of all legal moves for the current state. We consider moves with win rates in the top-3 as "top moves" for subsequent analysis. Additionally, we employed two supplementary engines as players in our testbed.

**Maia-1100**. To better understand the gap between the LLMs and human chess players, we incorporated Maia-1100 (McIlroy-Young et al., 2020), a chess AI with an Elo rating of approximately 1600 on real human chess platforms,[2] which is roughly the average level for human players. Maia-1100 is specially developed for chess, and it is based on CNN and Monte Carlo Tree Search.

---

[2]https://lichess.org/@/maia1

**Random Player**. We also included a random player, which chooses randomly from all legal moves on the board. Note that this player is not purely random, as we provide the legal moves to it.

## 2.5 FINE-GRAINED EVALUATION TASKS

In addition to the overall Glicko rating, ChessArena also provides more comprehensive evaluations of the strategic reasoning capabilities of LLMs. We design three fine-grained tasks as follows.

**Basic Understanding**. This task evaluates models' basic understanding of chess rules and board states by testing their ability to generate legal moves. Given a chessboard state and a specific position, models must identify the piece of the given position (e.g., King or Queen) and generate all legal moves of this piece. We assess this basic understanding capability using three metrics: *Piece Match Accuracy (PMA)*, which measures the accuracy of piece identification, and *Precision* and *Recall*, which measure the accuracy of legal move prediction. To strengthen the evaluation, we introduce perturbations including empty squares and turn-mismatch scenarios (e.g., requesting a Black piece when it is White's turn). In such cases, the correct response should be no legal moves.

**Move Selection**. This task evaluates models' single-move chess-playing ability by requiring them to select optimal moves from a given board state. We assess performance using three metrics: *Legal Rate (LR)*, *Top Rate (TR)*, and *Move Advantage Rate (MAR)*. LR quantifies the proportion of legal moves predicted by the model. TR evaluates whether the model's predictions are included in the "top moves" as determined by Stockfish. MAR measures the relative strength of a model's predicted move compared to all legal moves. Using Stockfish to evaluate win rates $Q(\text{FEN}, \text{move})$ for all legal *moves* from a given chessboard, we compute the Average Win Rate (AWR) as $\frac{1}{M} \sum_{m=1}^{M} Q(\text{FEN}, \text{Move}_m)$, where $M$ is the number of legal moves. MAR is then calculated as:

$$\text{MAR} = \frac{1}{N} \sum_{i=1}^{N} \frac{Q(\text{FEN}_i, \text{Move}_{\text{pred}}) - \text{AWR}_i}{\text{AWR}_i},$$

where $N$ is the number of evaluation instances. For illegal moves, we set $Q(\text{FEN}, \text{Move}_{\text{pred}}) = 0$.

**Puzzle Solving**. In line with the work of Hwang et al. (2025) and Ruoss et al. (2024), we evaluate chess puzzle solving using the Lichess puzzle database[3]. Each puzzle begins from an initial board state and consists of $k$ ground-truth sequential moves that represent the solution. At each step, we present the current board state and require the model to predict the optimal move. A puzzle is considered solved only if all $k$ predicted moves match the ground truth exactly–even a single error in any step results in failure. We utilize the puzzle dataset from the Lichess puzzle database, where each puzzle is associated with an Elo rating ranging from 200 to 3000 on the Lichess platform. We use *Puzzle Solving Accuracy (PSA)* as our evaluation metric, which measures the percentage of puzzles that the model correctly solves.

## 3 POST-TRAIN LLMS FOR CHESSARENA

To explore potential solutions to the observed strategic reasoning limitations exhibited by the models, we post-train LLMs (named Qwen3-8B-Chess and Seed-Coder-8B-Chess) on Qwen3-8B and Seed-Coder-8B-Instruct, which are the weakest among the studied LLMs. Our post-training includes two stages of supervised fine-tuning (SFT) and one stage of group relative policy optimization (GRPO) (Shao et al., 2024).

**Supervised fine-tuning**. This phase aims to gain basic chess reasoning ability. It consists of two stages. In the first stage of SFT, we use chess-based dialogue data from ChessGPT (Feng et al., 2023), which covers discussions on basic chess rules, tactics, etc. This stage injects the background knowledge about chess into the model. In the second stage of SFT, we collect and filter data from games played on ChessArena, which is critical for endowing the model with fundamental chess reasoning skills.

**Group relative policy optimization**. In the following stage, we further enhance the chess ability through GRPO. GRPO has been demonstrated as an effective method for enhancing a model's

---

[3]https://database.lichess.org

reasoning capabilities, particularly when verifiable rewards (e.g., for mathematics or code generation) are employed (Guo et al., 2025). This is also the case for our chess scenario. Specifically, we utilize Stockfish to analyze the model's moves and generate verifiable reward signals, enabling the model to autonomously explore chess strategies through this feedback mechanism. We define three types of rewards: format reward, legal move reward, and top move reward. For more details about post-training (e.g., training data collection, reward design, and training hyperparameters), please refer to Appendix D.

## 4 EXPERIMENTS

### 4.1 EXPERIMENTAL SETUP

**Evaluated Models**. We evaluated leading proprietary and open-source LLMs, including GPT-4.1 (OpenAI, 2025a), GPT-4o (OpenAI, 2024), O3 (OpenAI, 2025b), Claude-3-7-Sonnet (Anthropic, 2025), Gemini-2.5-pro (Google DeepMind, 2025), Qwen3-235B-A22B(Non-thinking) (Yang et al., 2025), DeepSeek-R1 (Guo et al., 2025), DeepSeek-V3 (Liu et al., 2024), DeepSeek-V3.1(Non-thinking), and Doubao (Seed et al., 2025) series. We also include Qwen3-8B and our trained Qwen3-8B-Chess. All evaluated models are shown in Table 5.

**Ranking System**. We initialized our Glicko rating system with $r = 1500$ and $d = 350$, setting a minimum rating deviation of $d = 50$ to ensure meaningful rating adjustments throughout the competition. Following Lichess, we display only players with $d \leq 100$, indicating statistically reliable ratings.

**Implementation Details**. We set the max new tokens to 4096 for non-thinking models and 16384 for thinking models. We set $temperature$ as 0.2 and $top\_p$ as 1 for all experiments, as we observed optimal model performance under these configurations. We evaluated the models under two distinct conditions: with and without the provision of legal moves. We provide the latest 10 moves as partial move history in UCI format to give models sufficient game context and prevent fivefold repetition draws that occurred frequently without this historical information. In ChessArena, the termination conditions adhere to standard chess rules, including checkmate, forfeit, stalemate, insufficient material, fivefold repetition, the 75-move rule, and move limit (please refer to Appendix B.3 for further details). Our experimental setup employs two deployment approaches: official model APIs for most models, and local deployment via vLLM (Kwon et al., 2023) for Qwen3-8B and its post-training versions.

For basic understanding and move selection evaluation, we constructed the evaluation data from actual competitions in ChessArena. There are 200 and 1000 samples in these two tasks, respectively. To ensure a fair comparison of the model's performance across the four play modes, we guaranteed that the evaluation data for these four play modes consisted of identical board states. For the Blindfold mode in the move selection experiment, the average number of conversation turns in our evaluation data is 47. For puzzle solving experiments, we use 1008 puzzles from the Lichess puzzle database. For more details about fine-grained evaluation data construction, please refer to Appendix E.

All of our prompt templates are shown in AppendixB.2, including prompt templates in ChessArena competitions of different play modes and fine-grained evaluation. In actual competitions, players compete in an even number of games (alternating between playing as White and Black) to balance the first-move advantage.

### 4.2 EXPERIMENTAL RESULTS

**(A) LeaderBoard**. Our rating leaderboard is shown in Table 1. We provide legal moves to the vast majority of models, as they cannot effectively play without them. The whole leaderboard and sub-metrics during competitions are shown in Appendix F.1.

*Key observations*. Among the models, we observe that thinking models such as O3, Doubao-Seed-1-6-Thinking, and Gemini-2.5-Pro currently lead the rankings. Among non-thinking models, GPT-4.1 and Claude-3-7-Sonnet show the strongest performance. When legal moves are provided, our trained Qwen3-8B-Chess achieves the best performance among all non-thinking models (including GPT-4.1). Notably, the untrained Qwen3-8B baseline ranks at the bottom of the leaderboard.

Table 1: Leaderboard of ChessArena. Thinking models generally perform better than non-thinking models, while all models are inferior to Maia-1100. When legal moves are provided, our post-trained Qwen3-8B-Chess outperforms other non-thinking models and is on par with thinking models. ('Legal Moves' stands for whether the set of legal moves are provided to the model, 'RD' means the rating deviation, and 'Interval' means the 95% confidence interval for the rating.)

| Rank | Model | Mode | Legal Moves | Rating | RD | Interval | Games |
|------|-------|------|-------------|--------|----|----------|-------|
| 1 | **Maia-1100** | - | × | 2220 | 82 | (2058, 2382) | 44 |
| 2 | O3 | Standard | × | 1948 | 78 | (1793, 2101) | 28 |
| 3 | Doubao-Seed-1-6-Thinking | Standard | ✓ | 1830 | 50 | (1729, 1929) | 60 |
| 4 | Gemini-2.5-Pro | Standard | ✓ | 1819 | 81 | (1659, 1979) | 18 |
| 5 | **Qwen3-8B-Chess** | Blitz | ✓ | 1776 | 93 | (1593, 1959) | 16 |
| 6 | Doubao-Seed-1-6-Thinking | Standard | × | 1743 | 66 | (1612, 1873) | 36 |
| 7 | GPT-4.1 | Blindfold | ✓ | 1699 | 50 | (1601, 1797) | 60 |
| 8 | Doubao-Seed-1-6-Thinking | Blindfold | ✓ | 1687 | 73 | (1542, 1831) | 24 |
| 9 | GPT-4.1 | Blitz | ✓ | 1686 | 50 | (1588, 1784) | 182 |
| 10 | Claude-3-7-Sonnet | Blitz | ✓ | 1654 | 50 | (1555, 1751) | 74 |
| 11 | Claude-3-7-Sonnet | Blindfold | ✓ | 1625 | 66 | (1493, 1756) | 30 |
| 12 | GPT-4.1 | Blitz | × | 1623 | 50 | (1525, 1721) | 106 |
| 13 | Gemini-2.5-Pro | Standard | × | 1616 | 74 | (1469, 1762) | 28 |
| 14 | Seed-Coder-8B-Chess | Blitz | ✓ | 1614 | 63 | (1490,1738) | 30 |
| 15 | Qwen3-8B-SFT-Stage2 | Blitz | ✓ | 1612 | 56 | (1501, 1721) | 40 |
| 16 | Claude-3-7-Sonnet | Blindfold | × | 1588 | 72 | (1445, 1729) | 28 |
| 17 | GPT-4.1 | Bullet | ✓ | 1583 | 50 | (1485, 1681) | 54 |
| 18 | DeepSeek-V3 | Blitz | ✓ | 1553 | 50 | (1454, 1650) | 174 |
| 19 | **Random Player** | - | ✓ | 1524 | 50 | (1425, 1621) | 284 |
| 20 | Qwen3-235B-A22B | Blitz | ✓ | 1483 | 50 | (1385, 1581) | 146 |
| 21 | DeepSeek-V3 | Blitz | × | 1482 | 58 | (1367, 1597) | 48 |
| 22 | DeepSeek-V3 | Blindfold | ✓ | 1437 | 75 | (1290, 1584) | 24 |
| 23 | DeepSeek-V3 | Bullet | ✓ | 1382 | 80 | (1224, 1540) | 22 |
| 24 | Qwen3-235B-A22B | Bullet | ✓ | 1369 | 54 | (1261, 1476) | 46 |
| 25 | Qwen3-8B | Blitz | ✓ | 1335 | 65 | (1205, 1463) | 32 |
| 26 | Seed-Coder-8B-Instruct | Blitz | ✓ | 1009 | 106 | (800,1218) | 30 |

*Comparing to Maia-1100 and Random Player.* There remains a significant performance gap between LLMs and Maia-1100. Currently, no LLM has demonstrated the capability to defeat Maia-1100 in actual gameplay, which demonstrates the inadequacy of the model's strategic reasoning capabilities. Compared to the random player baseline, most models exhibit better performance. However, a few models still underperform the random player. This primarily occurs when models fail to generate legal moves due to the lack of instruction-following ability, resulting in forfeit losses. Table 12 presents metrics demonstrating substantial *parsing error rates* across models, indicating format non-compliance and instruction-following deficiencies. Several models also exhibit elevated *illegal move rates* even when legal moves are explicitly provided. While high *illegal move rates* are anticipated when legal moves are not provided, rates exceeding 5% in scenarios with provided legal moves warrant attention.

*Different play modes.* For the same model under different play modes, we observe that most models achieve their best performance in Blitz or Standard modes. This aligns with expectations, as these modes provide the model with the most direct board information while permitting reasoning. Under Blindfold conditions, O3, Doubao-Seed-1-6-Thinking, GPT-4.1, and Claude-3-7-Sonnet still demonstrate competent playing strength. They demonstrate stronger multi-turn memorization and long-term strategic reasoning capabilities than other models. However, in Bullet mode, nearly all models perform poorly. This suggests that prohibiting thought chain output (e.g., "Let me think ..." or reasoning steps) severely impairs the models' chess strategic reasoning capabilities.

**(B) Basic Understanding**. Table 2 shows the results of the basic understanding task. It can be seen that thinking models (e.g., O3, Doubao-Seed-1-6-Thinking and DeepSeek-R1) have almost complete chessboard understanding capabilities, being able to identify pieces at specific positions on the board and generate related legal moves according to chess rules. Additionally, some strong non-thinking models, such as GPT-4.1 and Claude-3-7-Sonnet, also have relatively high *PMA*, *Precision* and *Recall*. Our trained Qwen3-8B-Chess shows improvement over Qwen3-8B on this task, even though we did not specifically train on this task.

Table 2: Basic understanding results. Thinking models such as O3 and Doubao-Seed-1-6-Thinking show strong chessboard understanding capabilities. Our post-training significantly improves the basic understanding capability.

| Model | PMA (%) | Precision (%) | Recall (%) |
|---|---|---|---|
| GPT-4.1 | 98.0 | 89.3 | 92.1 |
| O3 | 98.5 | 98.5 | 98.5 |
| DeepSeek-V3 | 97.0 | 81.8 | 75.3 |
| DeepSeek-V3.1 | 89.0 | 87.5 | 87.4 |
| DeepSeek-R1 | 100.0 | 99.2 | 98.4 |
| Doubao-1-5-Pro-32k | 76.0 | 50.6 | 56.2 |
| Doubao-1-5-Lite-32k | 51.5 | 33.3 | 30.3 |
| Doubao-1-5-Thinking-Pro | 99.5 | 98.0 | 98.0 |
| Doubao-Seed-1-6-Thinking | 100.0 | 99.9 | 99.9 |
| Qwen3-235B-A22B | 80.5 | 50.7 | 49.3 |
| Claude-3-7-Sonnet | 98.0 | 87.6 | 87.3 |
| Gemini-2.5-Pro | 100.0 | 98.5 | 96.7 |
| Qwen3-8B | 36.0 | 14.1 | 18.8 |
| Qwen3-8B-Chess-SFT-Stage1 | 63.5 (+31.5) | 20.6 (+5.9) | 29.5 (+14.3) |
| Qwen3-8B-Chess-SFT-Stage2 | 70.5 (+7.0) | 51.9 (+31.3) | 45.3 (+15.8) |
| Qwen3-8B-Chess (SFT+RL) | 79.0 (+8.5) | 52.6 (+0.7) | 50.1 (+4.8) |

**(C) Move Selection**. Table 3 shows the results of move selection. We share our findings below.

*LLMs have significant room for improvement in the strategic reasoning of chess.* Among all LLMs we evaluated, thinking models such as O3, Gemini-2.5-Pro, and Doubao-Seed-1-6-Thinking performed the best, while GPT-4.1 and Qwen3-8B-Chess also showed relatively excellent performance. However, their TP and MAR are far behind Maia-1100, which indicates that LLMs still have significant room for improvement in chess strategic reasoning capabilities. When legal moves are not provided, the performance of most models is even worse, as indicated by the negative MAR values.

*Bullet and Blindfold chess games bring difficulties to LLMs.* In terms of comparison among different play modes, the performance of LLMs (e.g, O3, Doubao-Seed-1-6-Thinking, DeepSeek-R1, GPT-4.1, Qwen3-235B-A22B, DeepSeek-V3.1) in Bullet or Blindfold mode is usually worse than in Blitz/Standard mode. Both Bullet (thinking content restricted) and Blindfold (multi-turn conversation reconstruction) pose certain difficulties for LLMs.

*Thinking models tried to reconstruct the chessboard.* For the Blindfold chess experiment, it appears that different models exhibit significant variations in performance. First of all, compared to Standard mode, DeepSeek-R1 and Doubao-Seed-1-6-thinking show a noticeable decline in performance in Blindfold mode. We manually checked their response and found that they were trying to reconstruct the chessboard, which brings much difficulty for them, especially when the number of conversation turns is large (i.e., more than 90 turns).

*Non-Thinking models may be lazy in Blindfold chess games.* GPT-4.1 and Qwen3-235B-A22B also demonstrate a significant drop when in Blindfold play mode compared to their performance in Blitz mode. For experiments Blindfold and without legal moves, we find that GPT-4.1, Qwen3-235B-A22B, and DeepSeek-V3 mostly base their responses on the last move in the conversation and continue from there, showing signs of laziness. They are more like guessing a move. Claude-3-7-Sonnet seems to reconstruct the chessboard genuinely and accomplishes this automatically without spending many response tokens. Overall, Blindfold chess poses significant challenges for models, revealing deficiencies in multi-turn reasoning capabilities.

Table 16 shows the average conversation turns for models predicting legal versus illegal moves in Blindfold mode without legal move provision. For non-thinking models, conversation turns had minimal impact on performance. However, thinking models required significantly fewer turns to predict legal moves compared to illegal ones, indicating that longer conversations impede board reconstruction. This disparity reveals the lazy behavior exhibited by non-thinking models. Notably, O3 maintains performance across more conversation turns than DeepSeek-R1 and Doubao-Seed-1.6-Thinking, demonstrating superior multi-turn memorization and reasoning capabilities. For more information about Blindfold mode analysis, please refer to Appendix G.2.

Table 3: Move selection performance across four play modes with/without legal moves provision. We bold the highest LR, TR, and MAR within each group. LLMs still have significant room for improvement, especially when the legal moves are not provided.

| Mode | Model or Engine | With Legal Moves | | | Without Legal Moves | | |
|------|-----------------|---------|---------|----------|---------|---------|----------|
| | | LR (%) | TR (%) | MAR (%) | LR (%) | TR (%) | MAR (%) |
| | Random Player | 100.0 | 14.8 | -1.1 | / | / | / |
| | Maia-1100 | / | / | / | 100.0 | 78.3 | +107.6 |
| **Blitz** | GPT-4.1 | 97.5 | 25.9 | +20.5 | 71.6 | **29.3** | **+6.2** |
| | Claude-3.7-Sonnet | **99.6** | 26.1 | +25.6 | 68.4 | 18.2 | -17.7 |
| | DeepSeek-V3 | 99.1 | 18.5 | +10.7 | 64.5 | 12.9 | -27.7 |
| | DeepSeek-V3.1 | 93.4 | 26.7 | +18.6 | 63.7 | 16.9 | -23.6 |
| | Qwen3-235B-A22B | 89.8 | 24.9 | +29.0 | 64.2 | 17.0 | -25.3 |
| | Qwen3-8B | 96.2 | 13.4 | +1.8 | 9.8 | 2.1 | -79.5 |
| | Qwen3-8B-Chess-SFT-Stage1 | 86.8 | 13.6 | -9.6 | 15.1 | 2.6 | -74.9 |
| | Qwen3-8B-Chess-SFT-Stage2 | 96.9 | 23.4 | +15.1 | 66.3 | 13.3 | -22.1 |
| | Qwen3-8B-Chess (SFT+RL) | 92.9 | **40.2** | **+41.1** | **87.6** | 20.2 | -1.2 |
| | Seed-Coder-8B-Instruct | 59.3 | 8.5 | -36.1 | 4.5 | 1.0 | -85.4 |
| | Seed-Coder-8B-Chess(SFT+RL) | 99.5 | 29.5 | +35.7 | 85.1 | 12.4 | -9.0 |
| **Bullet** | GPT-4.1 | 98.7 | **25.0** | **+20.8** | 74.0 | **28.7** | **+5.7** |
| | Claude-3.7-Sonnet | **98.6** | 22.5 | +16.8 | **75.2** | 17.9 | -9.4 |
| | DeepSeek-V3 | 98.9 | 18.8 | +11.3 | 66.2 | 13.3 | -21.8 |
| | DeepSeek-V3.1 | 80.6 | 16.1 | -8.0 | 56.3 | 12.7 | -35.7 |
| | Qwen3-235B-A22B | 95.9 | 17.8 | +4.5 | 69.1 | 15.9 | -18.5 |
| **Standard** | DeepSeek-R1 | **100.0** | 32.7 | +34.7 | 82.5 | 23.7 | -1.0 |
| | Doubao-1-5-Thinking-Pro | 99.7 | 32.9 | +35.4 | 78.0 | 24.8 | +3.0 |
| | Doubao-Seed-1-6-Thinking | 99.8 | 39.1 | +53.7 | 90.7 | 36.0 | +32.0 |
| | Gemini-2.5-Pro | 99.4 | 37.6 | +46.5 | 85.5 | 40.5 | +36.5 |
| | O3 | 99.6 | **58.7** | **+80.1** | **98.0** | **62.0** | **+80.2** |
| **Blindfold** | GPT-4.1 | 96.8 | 20.1 | +12.7 | 72.7 | 20.2 | +1.2 |
| | Claude-3.7-Sonnet | 98.2 | 23.9 | +21.5 | 77.3 | 18.9 | -9.1 |
| | DeepSeek-V3 | 95.1 | 19.2 | +16.2 | 78.5 | 14.9 | -7.8 |
| | DeepSeek-V3.1 | 96.5 | 26.0 | +27.2 | 66.0 | 13.7 | -18.0 |
| | DeepSeek-R1 | 94.7 | 22.7 | +14.0 | 44.6 | 10.9 | -36.9 |
| | Qwen3-235B-A22B | 96.1 | 19.9 | +17.4 | 75.3 | 17.2 | -10.4 |
| | Doubao-Seed-1-6-Thinking | 97.8 | 32.1 | +36.5 | 43.6 | 12.9 | -30.5 |
| | Gemini-2.5-Pro | 98.7 | 30.4 | +23.5 | 68.7 | 21.5 | -8.7 |
| | O3 | **98.4** | **46.9** | **+63.2** | 86.9 | 43.5 | +50.9 |

**(D) Puzzle solving**. Table 4 presents the main experimental results of puzzle solving, where we divide the puzzles according to their Elo ratings. Stockfish achieved an overall score of 98.4%, which aligns with expectations. Maia-1100 attained an overall score of 74.6%. Among all LLMs, the O3 model stands out remarkably, achieving a puzzle-solving rate of 55.6%. Other models all scored below 15%. Overall, thinking models outperformed non-thinking models. Our trained Qwen3-8B-Chess achieved the highest score among non-thinking models. We also present the puzzle-solving results without legal moves in Table 17. As can be observed, stronger models such as O3, GPT-4.1, and Gemini-2.5-pro exhibit almost no performance degradation, whereas weaker models are significantly affected. The models' deficiencies in puzzle solving task indicate persistent limitations in long-term reasoning capabilities.

## 5 RELATED WORK

**Chess Language Model**. Recent studies have explored LLM applications to chess with interesting findings. Xiangqi-R1 (Chen et al., 2025) achieved strong performance in Chinese chess through SFT and GRPO training, while Hwang et al. (2025) encountered significant bottlenecks when applying RL methods to chess puzzle solving, which the authors attribute to the model's inadequate acquisition of chess-related knowledge during pretraining. Carlini (2023) discovered that GPT-3.5-turbo could play chess using PGN format, but deeper analysis revealed reliance on memorized patterns rather than genuine reasoning. In contrast, Chess Bench (Ruoss et al., 2024) achieved grandmaster-level performance using a 270M Transformer pre-trained through Stockfish knowledge distillation, though this represents a domain-specific architecture rather than a general language model. ChessGPT (Feng

Table 4: Puzzle solving accuracy when legal moves are provided. LLMs perform relatively poorly, with O3 standing out as the strongest.

| Model or Engine | Puzzle Solving Accuracy (%) | | | | | | | |
|---|---|---|---|---|---|---|---|---|
| | 200-600 | 600-1000 | 1000-1400 | 1400-1800 | 1800-2200 | 2200-2600 | 2600-3000 | Overall |
| Stockfish (Depth=20) | 100.0 | 100.0 | 100.0 | 100.0 | 99.3 | 97.9 | 91.5 | **98.4** |
| Maia-1100 | 98.6 | 97.2 | 91.6 | 82.5 | 72.7 | 51.0 | 28.2 | 74.6 |
| Random Player | 1.4 | 1.4 | 2.1 | 0.0 | 0.0 | 0.0 | 0.0 | 0.7 |
| GPT-4.1 | 18.9 | 14.0 | 8.4 | 4.9 | 1.4 | 2.8 | 0.0 | 7.2 |
| Claude-3-7-Sonnet | 18.2 | 16.1 | 4.9 | 4.2 | 5.6 | 1.4 | 0.0 | 7.2 |
| DeepSeek-V3 | 11.9 | 7.7 | 2.1 | 0.7 | 0.0 | 0.7 | 0.0 | 3.3 |
| DeepSeek-V3.1 | 13.3 | 10.5 | 8.4 | 4.9 | 1.4 | 2.8 | 7.0 | 6.0 |
| Qwen3-235B-A22B | 24.5 | 18.2 | 9.8 | 5.6 | 4.2 | 1.4 | 0.0 | 9.1 |
| Qwen3-8B | 2.8 | 4.9 | 2.1 | 0.0 | 0.0 | 0.0 | 0.0 | 1.4 |
| Qwen3-8B-Chess | 31.5 | 16.8 | 10.5 | 7.0 | 5.6 | 2.1 | 0.0 | **10.5** |
| Seed-Coder-8B-Instruct | 0.0 | 1.4 | 0.0 | 0.0 | 1.0 | 0.0 | 0.0 | 0.4 |
| Seed-Coder-8B-Chess | 23.8 | 8.4 | 4.9 | 3.5 | 4.9 | 2.8 | 0.0 | 6.9 |
| O3 | 97.9 | 90.2 | 79.7 | 62.9 | 46.5 | 10.5 | 1.4 | **55.6** |
| Gemini-2.5-Pro | 37.1 | 24.5 | 18.2 | 9.1 | 4.2 | 3.5 | 1.4 | 14.0 |
| Doubao-Seed-1-6-Thinking | 27.3 | 23.8 | 11.9 | 7.7 | 4.2 | 1.4 | 2.1 | 11.2 |
| DeepSeek-R1 | 23.1 | 20.3 | 7.0 | 4.2 | 2.8 | 0.7 | 0.7 | 8.4 |

et al., 2023) represents a systematic approach, fine-tuning RedPajama-3B on web-scraped chess data to significantly outperform base models, while also contributing a high-quality chess-related training dataset. Similarly, Wang et al. (2025) fine-tuned LLaMA3-8B on expert-annotated datasets targeting tactics and strategy, achieving performance superior to GPT-4o on their benchmarks. However, their evaluation task—selecting the better move between two given options—is considerably less challenging than actual gameplay.

**LLMs Evaluation Benchmark**. ChatBot Arena (Chiang et al., 2024) introduced human preference-based evaluation using Elo rankings, grounding model assessment in naturalistic user interactions. SWE-Bench (Jimenez et al., 2023) evaluates LLMs on real-world software engineering tasks, while LiveCodeBench (Jain et al., 2024) provides continuously updated coding benchmarks from LeetCode and CodeForces to prevent data contamination. AIME2025 (MAA, 2025) assesses mathematical reasoning through 30 olympiad-level problems from the American Invitational Mathematics Examination. For strategic reasoning evaluation, GT-Bench (Duan et al., 2024) employs game-based scenarios to assess LLMs' strategic capabilities. ZebraLogic (Lin et al., 2025) tests logical reasoning through zebra puzzles of varying complexity. BBH (Kazemi et al., 2025) comprises 23 challenging multi-step reasoning tasks from BIG-Bench. Most relevant to our work, a concurrent work, GameArena (Lee et al., 2025), evaluated eight LLMs' chess abilities in blitz-style competitions. We introduce ChessArena as a comprehensive testbed with multiple gameplay scenarios and fine-grained studies to evaluate strategic reasoning capabilities in current language models.

## 6 DISCUSSION

**Generalization.** One interesting problem is whether models trained on chess-specific domains with enhanced strategic reasoning capabilities can be generalized to other domains. In Appendix G.3, we evaluate our chess-specific trained model's performance on other benchmarks. We found that our trained model demonstrated improvements on benchmarks such as AIME2025 and ZebraLogic, while maintaining comparable performance on other benchmarks.

**Limitations.** Our training data underwent outcome supervision filtering (evaluating only the quality of final moves without examining the reasoning process); this may result in training data containing cases where the reasoning process is flawed, but the final move is correct, potentially introducing noise into the dataset. This is a common issue shared across domains that rely on outcome supervision, such as code generation and mathematical reasoning. Besides, our trained model performs well under the "with legal moves" setting but poorly when such legal moves are not provided. This indicates that the model may still depend on memorization instead of developing genuine strategic understanding—a challenge potentially too demanding for 8B-parameter models. Alternatively, employing continued pre-training (Zhou et al., 2024) for the first stage of our SFT may be a viable option to improve its capability.

**Conclusion.** We introduce ChessArena, a competitive platform enabling large language models to play against each other in human-like chess competitions. Through authentic gameplay, we evaluate LLMs' strategic reasoning, instruction following, and multi-turn conversational memorization capabilities. Our analysis through ChessArena gameplay and fine-grained evaluation reveals substantial room for improvement in LLMs' chess strategic reasoning abilities. Observing deficiencies of current LLMs, we trained Qwen3-8B-Chess and achieved significant improvements in chess strategic reasoning capabilities. We hope our ChessArena platform, fine-grained evaluation tasks, and high-quality training datasets will contribute to future large language model research.

ETHICS STATEMENT

ChessArena is constructed entirely from publicly available chess gameplay data and standard chess notation formats that permit open research usage, ensuring our contributions comply with established data usage protocols. During the collection and evaluation processes, we do not gather personal information about players or participants, and ChessArena task instances utilize only standard chess positions and moves that are part of the public domain of chess knowledge. Our contributions do not involve any human subject participation; we do not perform crowdsourcing or recruit human annotators for any component of ChessArena, including data collection, game execution, and evaluation procedures. ChessArena's model selection and evaluation criteria are based solely on objective performance metrics and do not implicitly or explicitly rely on any discriminative or biased heuristics for model assessment. For the dataset release, we plan to open-source the ChessArena competition platform, the ranking system implementation, competition sampling algorithms, fine-grained evaluation frameworks, experimental results, the training data used for fine-tuning the Qwen3-8B-Chess model, and the model weights. Following established best practices, we will provide comprehensive documentation describing each component and its usage, and establish accessible communication channels for soliciting community feedback to improve ChessArena. ChessArena does not present any immediately harmful applications, as chess gameplay represents a benign domain for evaluating strategic reasoning capabilities.

REPRODUCIBILITY STATEMENT

In our submitted source code, we provide all corresponding code for ChessArena competition, ranking system, competition sampling, and fine-grained evaluation, as well as chess training-related SFT and GRPO training data examples and code, enabling readers to reproduce our results. In the future, we plan to open-source all code with corresponding documentation. We also intend to release the trained Qwen3-8B-Chess model to facilitate subsequent research.

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

APPENDIX

In the Appendix, we provide detailed experimental settings, mathematical proofs for the competition sampling algorithm, post-training dataset construction and training details, and additional experimental results and analysis.

## A    THE USAGE OF LARGE LANGUAGE MODELS

We used large language models as assistant tools for two specific purposes in this work:

- Paper polishing: LLMs were employed to improve the clarity, grammar, and overall readability of the manuscript text.
- Table formatting: LLMs assisted in enhancing the visual presentation and formatting of tables to improve readability.

The use of LLMs was limited strictly to these two auxiliary functions. LLMs did not contribute to research ideation, experimental design, data analysis, result interpretation, or the generation of scientific content. All research ideas, methodologies, findings, and conclusions are entirely the work of the authors.

## B    MORE IMPLEMENTATION DETAILS

### B.1    EVALUATED MODELS

Our evaluated models are shown in Table 5.

Table 5: Large Language Models Evaluated in ChessArena

| Model Family | Model Name | Type | Thinking |
|---|---|---|---|
| **OpenAI** | GPT-4.1 (OpenAI, 2025a) | Proprietary | ✗ |
| | GPT-4o (OpenAI, 2024) | Proprietary | ✗ |
| | O3 (2025-04-16) (OpenAI, 2025b) | Proprietary | ✓ |
| **DeepSeek** | DeepSeek-V3 (0324) (Liu et al., 2024) | Open Source | ✗ |
| | DeepSeek-V3.1 | Open Source | ✗ |
| | DeepSeek-R1 (0120) (Guo et al., 2025) | Open Source | ✓ |
| **ByteDance** | Doubao-1.5-Pro-32K | Proprietary | ✗ |
| | Doubao-1.5-Lite-32K | Proprietary | ✗ |
| | Seed-Coder-8B-Instruct | Open Source | ✗ |
| | Doubao-1.5-Thinking-Pro | Proprietary | ✓ |
| | Doubao-Seed-1.6-Thinking (Seed et al., 2025) | Proprietary | ✓ |
| **Alibaba** | Qwen3-235B-A22B (0514) (Yang et al., 2025) | Open Source | ✗ |
| | Qwen3-8B (0514) | Open Source | ✗ |
| **Anthropic** | Claude-3.7-Sonnet (Anthropic, 2025) | Proprietary | ✗ |
| **Google** | Gemini-2.5-Pro (Google DeepMind, 2025) | Proprietary | ✓ |
| **Ours** | Qwen3-8B-SFT-Stage1 | Open Source | ✗ |
| | Qwen3-8B-SFT-Stage2 | Open Source | ✗ |
| | Qwen3-8B-Chess (SFT+RL) | Open Source | ✗ |
| | Seed-Coder-8B-SFT-Stage1 | Open Source | ✗ |
| | Seed-Coder-8B-SFT-Stage2 | Open Source | ✗ |
| | Seed-Coder-8B-Chess(SFT + RL) | Open Source | ✗ |

### B.2    PROMPT TEMPLATES

There are the prompt templates for chess competitions, designed for various play modes. Blitz and Standard allow the model to think, as shown in Figure 2. Bullet expects the model to output the answer directly without thinking, as shown in Figure 3. Blindfold is another mode, where the model is expected to reconstruct the board from the conversation history and play accordingly. We record the player's and opponent's moves in the conversation history. For details, please refer to Figure 4.

For basic understanding, the prompt templates are shown in Figure 5. The prompt templates for move selection remain consistent with those of each play mode. The prompt template of puzzle solving is the same as Blitz/Standard play mode prompt template.

## B.3 TERMINATION CONDITIONS

Following official chess rules, our games terminate under these conditions:

- **Checkmate**: A player delivers checkmate, winning the game.

- **Forfeit**: A player fails to generate a legal move after multiple attempts (indicating either instruction-following deficits or board analysis failures), awarding victory to the opponent. We will give an instruction if an LLM fails to give a legal move. We allow a player to retry 5 times.

- **Stalemate**: Draw declared when the active player has no legal moves but is not in check.

- **Insufficient material**: Draw due to neither player having sufficient pieces to force checkmate (e.g., king vs. king).

- **Fivefold repetition**: Draw triggered by the same position recurring five times.

- **75-move rule**: Draw if 75 consecutive moves occur without pawn advances or captures.

- **Move limit**: Draw if the total move count exceeds the maximum move count. We set it to 200 moves.

## B.4 CHESS NOTATION

**Board Representation**. We adopt the Forsyth-Edwards Notation (FEN) rec.games.chess (1994) as our chessboard representation Standard. FEN is a widely recognized notation system that encodes a chess position into six space-delimited fields, comprehensively capturing the game state (e.g., piece placement, active color, castling rights, en passant targets, half move clock, and full move number). This notation is supported by the Python-Chess library and provides LLMs with an unambiguous, machine-parseable representation of board states, where each unique chess position maps to a distinct FEN string.

**Move Representation**. For move encoding, we implement the Universal Chess Interface (UCI) Kahlen (2004) Standard, which specifies moves in coordinate notation (e.g., "e2e4" for pawn advance). UCI's start-to-end positional format ensures deterministic move interpretation. Additionally, we maintain compatibility with Standard Algebraic Notation (SAN) to accommodate alternative LLM outputs. Our system automatically normalizes all move representations into a canonical form, enabling robust analysis regardless of the LLM's native output format.

In ChessArena, we first prefer to have models output UCI notation moves. If UCI notation moves cannot be extracted, we will extract SAN moves. We support both move notations. Regarding chessboard representations, we know that besides FEN representation, there is also Portable Game Notation (PGN) rec.games.chess (1994) representation. However, PGN representation shows the move history of a game and cannot directly reveal the piece arrangement on a board, so we use FEN representation, which is much more direct and clear for models. In basic understanding experiments, GPT-4.1, Claude-3-7-sonnet, and Doubao-Seed-1-6-thinking all showed a high piece match rate, precision, and recall, indicating they have understanding capabilities for FEN board representation, but their actual chess gameplay performance still has considerable space for improvement.

## B.5 DIFFERENCE BETWEEN MOVE SELECTION AND REAL CHESS COMPETITION

Move selection evaluation results show an overall consistent trend with the ChessArena Leaderboard. ChessArena competition is more complex than single move selection. Additionally, LLMs also have opportunities to adjust themselves. LLMs must try their best in a pressure situation. Move selection offers a straightforward and efficient method for assessing an LLM's chess strategic reasoning. In contrast, ChessArena competition provides a more accurate and engaging evaluation by requiring models to participate in extended game sessions.

**Blitz/Standard prompt template**

**System:**
You are an expert chess player.You are playing a game
of chess.You are playing as {White_or_Black}.
You must thoroughly analyze the position and play with
utmost caution.When you have the advantage, press it
relentlessly and aim for a swift checkmate.Carefully
evaluate every move to eliminate any chance of a
counterplay or draw by your opponent.
When at a disadvantage, strive to turn the tide and
win if possible.If victory is unattainable, exhaust
all possible means to force a draw.
Meticulously analyze legal moves, then select the
absolute best one.  You need to determine whether
you are playing as Black or White.Then, you need to
observe the positions of your pieces and choose one
of your own pieces to move; make sure that your move
follows the rules of chess.
Considering the long-term strategy and short-term
tactic.Analyze the position carefully.You may think
through the position and consider multiple candidate
moves.
When you have decided on your final move, output it in
UCI notation (e.g., 'e2e4', 'g8f6' , 'e7e8q') in the
following format:
<answer>
```
<move>
```
</answer>
For example:
```
e2e4
```

Reminder of chess rules:
    - Bishops move diagonally.
    - Rooks move horizontally or vertically.
    - Knights jump in an L-shape.
    - Queens combine rook and bishop movement.
    - Kings move one square in any direction.
    - Pawns move forward, capture diagonally, and can
promote.
You can think and reason as much as you want(step by
step), but your final move must be formatted exactly
as shown above.
**User:**
The current FEN: {<fen>}
(Optional) Legal moves in UCI notation:  {<uci_1>
<uci_2> ...<uci_L>}.
What is the best move?

Figure 2: Input prompt format for Blitz and Standard chess competition. Whether to provide legal moves is optional.

---

**Bullet prompt template**

**System:**
You are an expert chess player.You are playing a game
of chess.You are playing as White/Black.
Same as Blitz. Omitted.
You must give me your answer **directly without using
any other words**.I will not accept your answer if there
are any other words.Only output your move content.Your
final move must be formatted exactly as shown above.

**User:**
The current FEN:{<fen>}
(Optional) Legal moves in UCI notation: {<uci_1>
<uci_2> ...<uci_L>}.
What is the best move?

---

Figure 3: Input prompt format for Bullet chess competition. Whether to provide legal moves is optional. Thinking is forbidden.

---

**Blindfold prompt template**

**System:**
You are an expert chess player.You are playing a game
of chess.You are playing as White/Black.
**We have the move history of you
and your opponent.You must reconstruct the game and
analyze the best move on the chessboard.**
Same as Blitz. Omitted.
You can think and reason as much as you want(step by
step), but your final move must be formatted exactly
as shown above.

**User(First Turn):**
This is the beginning of the game.

**Assistant(First Turn):**
```
<move_1>
```

**User:**
Your **opponent's last move** is <move_2>.

**Assitant:**
```
<move_3>
```

Multi-Turns
**User:** Your **opponent's last move** is {<move_k>}.
(Optional) Legal moves in UCI notation: {<uci_1>
<uci_2> ...<uci_L>}.
What is the best move?

---

Figure 4: Input prompt format for Blindfold chess competition. Whether to provide legal moves is optional. This is a multi-round conversation template. LLMs should reconstruct the chessboard from the conversation history.

**Basic understanding evaluation prompt template**

**System:**
You are an expert chess player.I need you to help me model a chessboard.The specific steps are as follows:
I will provide you with a FEN string representing the current board state,and then give you a position.You need to identify the piece at that position from the FEN and output all legal moves for that piece.
You must carefully analyze the board, consider the rules of chess, and provide the final answer.

Your answer should be format as follows(output a json):
```json
{
    "piece": <piece symbol>,
    "legal moves": [<list of legal moves>]
}
```

For example:
FEN: rnbqkbnr/pppppppp/8/8/8/8/PPPPPPPP/RNBQKBNR w KQkq - 0 1
Position:g1
Answer:
```json
{
    "piece": "N",
    "legal moves": ["g1h3", "g1f3"]
}
```
Note:
If the given position has no piece, directly output empty(i.e.,None), and the corresponding legal moves should also be empty(i.e.,[]).
When it's White's turn to move, if the position contains a Black piece, you should identify the piece, but its legal moves must be empty (and vice versa for Black's turn).
You can think and reason as much as you want(step by step), but your final answer must be formatted exactly as shown above.
**User:**
Current board position in FEN notation:{<fen>}
Position:{<pos>}

Figure 5: Input prompt format for basic understanding

## C  GLICKO RATING SYSTEM & COMPETITION SAMPLING ALGORITHM

### C.1  GLICKO RATING SYSTEM

In the Glicko ranking system, each player is assigned two values: the rating $r$ and the rating deviation RD. In the original Glicko paper Glickman (1995), these values are updated after a certain period; In our scenario, we assume they are updated after each competition. The updated values, denoted as $r'$ and $RD'$, are given by the following formulas:

$$r' = r + \frac{q}{\frac{1}{\text{RD}^2} + \frac{1}{d^2}} g(\text{RD}_o) \left(s_o - \text{E}(s \mid r, r_o, \text{RD}_o)\right) \tag{1}$$

$$\text{RD}' = \sqrt{\left(\frac{1}{\text{RD}^2} + \frac{1}{d^2}\right)^{-1}} \tag{2}$$

where

$$q = \frac{\ln 10}{400} \approx 0.0057565 \tag{3}$$

$$g(\text{RD}) = \frac{1}{\sqrt{1 + \frac{3q^2 \text{RD}^2}{\pi^2}}} \tag{4}$$

$$\text{E}(s \mid r, r_o, \text{RD}_o) = \frac{1}{1 + 10^{-g(\text{RD}_o)(r - r_o)/400}} \tag{5}$$

$$d^2 = \left(q^2 \left(g(\text{RD}_o)\right)^2 \text{E}(s \mid r, r_o, \text{RD}_o) \left(1 - \text{E}(s \mid r, r_o, \text{RD}_o)\right)\right)^{-1} \tag{6}$$

where $s$ represents the competition result (i.e., 1 for a win, 0.5 for a draw, and 0 for a loss), $r_o$ and $RD_o$ denote the opponent's rating and rating deviation, respectively. These calculations are performed for each player participating in the rating period.

### C.2  COMPETITION SAMPLING

**System Objectives and Optimization Criteria**   In equation equation 2, we can see that RD will definitely decrease as matches progress, indicating that a player's rating becomes increasingly reliable. So under what conditions does a player's RD decay faster, enabling the player to converge most quickly? We provide a mathematical analysis in this section. The core goal of this matching system is to accelerate the convergence rate of player ratings, specifically by maximizing the reduction rate of rating deviation (RD). The optimization objective function is defined as:

$$\arg \max_{r_i, r_j, \text{RD}_i, \text{RD}_j} \Delta \text{RD}_i + \Delta \text{RD}_j \tag{7}$$

where $\Delta \text{RD}_i$ and $\Delta \text{RD}_j$ represent the changes in rating deviation for player $i$ and player $j$ after matching, respectively.

**Mathematical Derivation Process**   According to the update rules of the Glicko-1 system, the change in rating deviation can be expressed as:

$$\Delta \text{RD}_i = \text{RD}_i - \sqrt{\left(\frac{1}{\text{RD}_i^2} + \frac{1}{d_i^2}\right)^{-1}} = \text{RD}_i - \sqrt{\frac{\text{RD}_i^2}{1 + \frac{\text{RD}_i^2}{d_i^2}}} \tag{8}$$

$$\Delta\text{RD}_j = \text{RD}_j - \sqrt{\left(\frac{1}{\text{RD}_j^2} + \frac{1}{d_j^2}\right)^{-1}} = \text{RD}_j - \sqrt{\frac{\text{RD}_j^2}{1 + \frac{\text{RD}_j^2}{d_j^2}}} \tag{9}$$

To maximize $\Delta\text{RD}_i + \Delta\text{RD}_j$, we need to minimize $d_i^2$ and $d_j^2$. According to equation equation 6:

$$d_i^2 = \left(q^2 \left(g(\text{RD}_j)\right)^2 \text{E}(s \mid r_i, r_j, \text{RD}_j)\left(1 - \text{E}(s \mid r_i, r_j, \text{RD}_j)\right)\right)^{-1} \tag{10}$$

Therefore, minimizing $d_i^2$ and $d_j^2$ is equivalent to maximizing: $q^2 g(\text{RD}_i)^2 E_i(1 - E_i)$ and $q^2 g(\text{RD}_j)^2 E_j(1 - E_j)$, where:

$$E_i = \frac{1}{1 + 10^{-g(\text{RD}_j)(r_i - r_j)/400}}, \quad E_i = 1 - E_j \tag{11}$$

$$g(\text{RD}) = \frac{1}{\sqrt{1 + \frac{3q^2\text{RD}^2}{\pi^2}}}, \quad q = \frac{\ln 10}{400} \approx 0.0057565 \tag{12}$$

Based on the above derivation, the optimization objective can be transformed into:

$$\texttt{argmax}_{r_i, r_j, \text{RD}_i, \text{RD}_j} E_i(1 - E_i)\left[g(\text{RD}_i)^2 + g(\text{RD}_j)^2\right] \tag{13}$$

**Key Conclusions** From equation (7), we can draw the following important conclusions: When $r_i = r_j$ (i.e., the two players have the same rating), then $E_i = E_j = 0.5$, at which point $E_i(1 - E_i)$ reaches its maximum value of 0.25. Meanwhile, $g(\text{RD})$ is a decreasing function of RD, meaning that smaller RD results in larger $g(\text{RD})$.

Therefore, the optimal matching strategy is:

• Prioritize matching players with similar ratings ($r_i \approx r_j$)

• Under the premise of similar ratings, select players with smaller rating deviations (RD)

This strategy ensures maximum information gain for both players in the competition, thereby accelerating rating convergence.

**Algorithm Premises** A minimum rating deviation threshold min_RD (a hyperparameter) is set. When a player's RD $\leq$ min_RD, their rating deviation no longer decreases.

**Competition Sampling Process**

1. A player initiates a match request, and the system records their current rating $r$ and rating deviation RD

2. The system searches for potential opponents in the match pool and calculates the matching score:

$$\text{score}(i, j) = E_i(1 - E_i)\left[g(\text{RD}_i)^2 + g(\text{RD}_j)^2\right]$$

3. The opponent with the highest matching score is prioritized

4. For players with high RD, the system prioritizes matching them with opponents who have low RD and similar ratings

5. After the opponent accepts the match, the match begins

6. After the match, both players' $r$ and RD are updated based on the results

**ChessArena Matching System Variants**   The system supports two startup modes:

1. Random startup mode:

   (a) A player is randomly selected from the player pool

   (b) The selected player automatically initiates a match request

   (c) Steps 2-6 of the Competition Sampling process are executed

2. Specified startup mode:

   (a) An initial player is specified by a human

   (b) The specified player initiates a match request

   (c) Steps 2-6 of the Competition Sampling process are executed

# D   POST-TRAINING DETAILS

## D.1   SFT DATA COLLECTION

**ChessGPT**   ChessGPT Feng et al. (2023) has open-sourced a text pre-training dataset and a post-training SFT dataset related to chess. These datasets include conversational data about chess, covering topics such as basic rules and tactical discussions. We sampled chess-related portions (GPT-4-Chess, Chess-Forums, and Chess-Modeling) from this dataset as part of our SFT data.

**Distillation**   We distilled data from non-thinking models: GPT-4.1, DeepSeek-V3, Qwen3-235b-a22b, and Claude-3-7-Sonnet; Thinking models: Doubao-Seed-1-6-thinking and DeepSeek-R1. The input prompt format resembles the Blitz play mode prompt template, and the output includes the model's analysis of the chessboard and the final move selection. We used Stockfish to ensure the quality of the distilled data. We only retained data where the final move was among the top three moves analyzed by Stockfish. We filtered the data whose response length is less than 100. The characteristics of the distilled dataset are shown in the Table 6. We use the tiktoken(cl100k-base)[4] tokenizer to estimate the length of the distilled dataset.

What's more, in ChessArena, LLMs may initially fail to provide a legal move in the first round but correct themselves in subsequent attempts. We also extract such data for training, as it helps the model learn multi-turn correction capabilities. There are 652 samples in the multi-turn correction dataset.

Table 6: Characteristics of the Main Distilled Dataset(excluding multi-turn correction data)

| Type | Count | Prompt Length (avg.) | Resp. Len. (avg.) | TOP1 | TOP2–3 |
|---|---|---|---|---|---|
| Non-Thinking | 21,278 | 575 | 527 | 10,273 | 11,005 |
| Thinking | 3,399 | 582 | 5,014 | 1,862 | 1,537 |
| Multi-Round Correction | 652 | 1693 | 343 | 276 | 376 |

Table 7: SFT data summary

| Dataset | | Count | Description | Prompt Length (avg.) | Resp. Len. (avg.) |
|---|---|---|---|---|---|
| ChessGPT(Stage1) | GPT4-Chess | 3908 | Chess-related synthesized data from GPT-4 | 41 | 38 |
| | Chess Forums | 5395 | Chess-related dialogues data from online platform | 245 | 178 |
| | Chess Modeling | 3000 | Chessboard understanding data like PGN to FEN, FEN to UCI et al. | 116 | 65 |
| Distilled Data(Stage2) | Move Selection | 21278 | Distilled single turn data | 524 | 527 |
| | Multi-Round Correction | 652 | Multi-Round Correction data | 1693 | 343 |

---

[4]https://cookbook.openai.com/examples/how_to_count_tokens_with_tiktoken

## D.2 RL Data Collection

Theoretically, our RL training data is virtually unlimited, as only a single chessboard state is required to conduct RL training. Accordingly, we extracted board state data from the Lichess database and constructed our RL training dataset. All board states corresponding to FEN positions used in Fine-Grained studies were filtered out, resulting in a final set of 56,000 training samples. We ensured a balanced distribution of board states across the opening games, middle games, and end games to facilitate comprehensive learning by the model. Although experiments with larger datasets were attempted, no further improvement in model performance was observed.

## D.3 Reward

Our reward function consists of three components: 1) Format reward; 2) Legal move reward; 3) Top move reward. Format reward guides the model to follow certain formats; Legal move reward guides the model to infer legal moves from the chessboard; Top move reward guides the model to acquire chess strategy reasoning capabilities. Top moves are analyzed by Stockfish, and we pre-process to obtain top moves before training to avoid Stockfish consuming excessive CPU resources during training.

**Format Reward**    If the model's output follows the specified format (i.e., it is contained within the prescribed block), the format reward $reward_f = 1$; otherwise, $reward_f = 0$.

**Legal Move Reward**    If the model's predicted move is among the legal moves for the current board position, the legal moves reward $reward_l = 1$; otherwise, $reward_l = 0$.

**Top Move Reward**    If the model's predicted move matches one of the top moves pre-analyzed by Stockfish, the top moves reward $reward_t = 1$; otherwise, $reward_t = 0$.

The final reward is calculated as a weighted sum of these three rewards:

$$Reward = \epsilon_f \times reward_f + \epsilon_l \times reward_l + \epsilon_t \times reward_t$$

where $\epsilon_f$, $\epsilon_l$, and $\epsilon_t$ are the corresponding weight coefficients.

## D.4 Training hyper-parameters

**Supervised Fine-tuning**    We train our models using the LlamaFactory Zheng et al. (2024) framework. Our hyper-parameters are shown in Table 8. In our training process, to utilize more data (though mixing thinking and non-thinking data for training may cause issues), we use the non-thinking distilled dataset to train our models. So, our models is trained in non-thinking mode. For the multi-turn correction dataset, we only train our model on the final turn response.

**Reinforcement Learning**    We train our models using the verl (Sheng et al., 2025) framework. All of our training experiments are finished on 8 NVIDIA H800 80GB GPUs. A single training experiment takes approximately 60 hours. To enhance model performance, we incorporated methodologies from DR. GRPO Liu et al. (2025), Reinforce++ Hu (2025), and DAPO Yu et al. (2025). It was observed that the integration of these techniques contributed to improved model performance.

## D.5 Continuous Reward

We experimented with a continuous reward function. Building upon the same SFT model, we trained it using 10k chess RL samples. Our proposed continuous reward is:

$$Reward = 1 - \frac{Rank_{move}}{len(legal_{moves})} + \epsilon_f \times reward_f$$

where $\epsilon_f = 0.1$. In simple terms, the worst-ranked legal move receives a reward of 0, and the top-ranked move receives a reward of 1. Additionally, a format reward is added for valid moves. The performance of the model on the single-step move selection task is shown in Table 9.

Table 8: Training hyper-parameters for Post-training.

| Model Training Hyper-parameters | | |
|---|---|---|

**Supervised Fine-tuning Stage 1**

| Hyperparameter | Value |
|---|---|
| Training Steps | 1038 |
| Optimizer | AdamW |
| Learning Rate | 5e-6 |
| Global Batch Size | 32 |
| Epochs | 3 |
| Warmup Ratio | 0.1 |
| Lr scheduler type | Cosine |
| Max tokens | 4096 |

**Supervised Fine-tuning Stage 2**

| Hyperparameter | Value |
|---|---|
| Training Steps | 2130 |
| Optimizer | AdamW |
| Learning Rate | 5e-6 |
| Global Batch Size | 32 |
| Epochs | 3 |
| Warmup Ratio | 0.1 |
| Lr scheduler type | Cosine |
| Max tokens | 4096 |

**Reinforcement Learning**

| Hyperparameter | Value |
|---|---|
| **Training Configuration** | |
| Training Steps | 1750 |
| Optimizer | AdamW |
| Learning Rate | 1e-6 |
| Global Batch Size | 128 |
| Mini Batch Size | 64 |
| Epochs | 4 |
| Max Tokens | 4096 |
| **GRPO Configuration** | |
| Number of Rollouts | 8 |
| Rollout Temperature | 1.0 |
| Rollout Topp | 1.0 |
| KL Loss Coefficient | 0 |
| Entropy Coefficient | 0 |
| Clip High | 0.28 |
| Clip Low | 0.2 |
| **Reward Configuration** | |
| $\epsilon_f$ | 0.1 |
| $\epsilon_l$ | 0.3 |
| $\epsilon_t$ | 0.6 |

Table 9: Performance of Qwen3-8B-Chess with continuous reward. The model trained with the continuous reward function performs significantly worse than the model trained with the discrete reward.

| | LR (%) | TR (%) | MAR (%) |
|---|---|---|---|
| With Legal Moves | 90.3 | 12.7 | -9.8 |
| Without Legal Moves | 84.0 | 10.8 | -14.7 |

The results indicate that the model trained with the continuous reward performs poorly, significantly trailing the discrete reward model. We hypothesize that in the game of chess, only learning the few best moves is effective. Making the model learn the relative ranking between all legal moves might be unnecessary and could even lead to reward hacking (where the model might not fully commit to exploring the very best moves).

### D.6 WHY DO WE CHOOSE SINGLE-STEP RL?

Chess is a sequential game, and intuitively, employing sequence-like multi-step Reinforcement Learning (RL) methods, such as self-play, seems more appropriate. However, this study adopts single-step RL because: the Stockfish analysis, which we use as the oracle, searches through subsequent multiple moves (we set the depth to 20). This implies that the optimal move for the current board state (derived from Stockfish analysis) has already considered many future steps; it is not limited to the immediate next move. While multi-step RL methods, such as self-play, might align more intuitively with the nature of Chess RL, resource limitations prevented us from conducting long-context post-training, thus leading us to primarily adopt single-step RL for training. Readers who are interested are encouraged to further explore the effectiveness of Chess training using methods similar to self-play.

## E   Fine-Grained Evaluation Dataset Construction

We extracted the FEN of board states that actually occurred, thereby minimizing the risk of data contamination to the greatest extent possible.

**ChessBoard Extraction**   In each game of ChessArena, numerous board states are generated (averaging 40 chessboard states per game). However, many of these board states may be duplicated, particularly in the opening phase. We extract distinct board states from this data while ensuring a balanced distribution across the opening-game, middle-game, and end-game stages (with the middle-game slightly outnumbering both the opening and endgame phases). A total of 79,441 FENs are collected. FENs already present in the training dataset are removed to prevent data contamination of the trained models. After this filtering, 57,511 FENs remain. These retained chessboard states can be utilized to construct the subsequent evaluation set.

**Basic Understanding**   The FEN data acquired from the Chessboard Extraction step is subsequently utilized for further dataset construction. In the basic understanding evaluation dataset construction process, each position is sampled according to the following distribution: with an 85% probability, a position containing one of the player's own pieces is selected; with a 7% probability, a position from the opponent's pieces is chosen; and with an 8% probability, an empty square is selected. Each board FEN is used to construct one basic understanding data instance, resulting in a total of 57,511 instances. To facilitate efficient evaluation, a subset of 200 instances is selected for assessment. Consistent experimental outcomes have been observed across subsets of varying sizes, including 200, 500, 1000, and larger.

**Move Selection**   Following the acquisition of the FEN board state data from the initial step, theoretically, all instances could serve as evaluation data for move selection. However, to ensure both accuracy and efficiency in the evaluation process, we performed a rollout using Qwen3-8B-Chess on each data instance. This procedure ensured a balanced distribution of easy, medium, and difficult problems. From this processed set, 1,000 instances were selected to constitute the evaluation dataset for the move selection fine-grained experiment. For Blindfold play mode, we use the real move history that happened in ChessArena as the conversation history. However, it is important to note that a comprehensive evaluation of models' chess capabilities can be achieved without relying on our provided FEN board representations, for instance, by extracting board states from the Lichess database. Furthermore, given the vast search space of chess and the virtually infinite number of possible board configurations, the risk of data contamination is negligible.

**Puzzle Solving**   We retrieved puzzle data from the Lichess database and randomly selected 1,008 samples to form the evaluation set. Subsequently, we partitioned the data into seven segments based on Elo rating intervals of 400 points, with each segment containing exactly 144 puzzle instances. The dataset is sufficiently large to allow discernible observation of the differences in puzzle-solving capabilities among the LLMs.

**Datasets Distribution**   As mentioned before, we performed a preemptive rollout procedure using Qwen3-8B-Chess on the move selection evaluation dataset to categorize the difficulty levels. The rollout was conducted 8 times per instance with hyper-parameters set to temperature 1.0 and top-p 0.95. We defined an instance as easy if Qwen3-8B-Chess selected a top-3 move in all 6-8 rollouts, medium if it did so in 3-5 rollouts, and hard if it never selected a top-3 move. The overall difficulty distribution and statistics of fine-grained evaluation are illustrated in Table 10.

## F   Additional Results

### F.1   Whole Leaderboard

Our complete rating leaderboard is presented in Table 11, which includes additional models that do not affect the conclusion analysis, as well as models with RD values exceeding 100. Table 12 reports secondary metrics from ChessArena competitions, encompassing win-loss number, instruction following metrics (parsing errors, forbidden moves, legal moves), and move quality measures (top

Table 10: Statistics of fine-grained evaluation datasets.

| Task | Category | Count | Percentage (%) |
|------|----------|-------|----------------|
| **Basic Understanding** | Normal Positions | 144 | 72.0 |
| | Empty Positions | 19 | 9.5 |
| | Opponent Positions | 37 | 18.5 |
| | *Total* | *200* | *100.0* |
| **Move Selection** | *Phase - Early (0-20)* | 241 | 24.1 |
| | *Phase - Middle (20-60)* | 472 | 47.2 |
| | *Phase - Late (>60)* | 287 | 28.7 |
| | *Difficulty - Easy* | 215 | 21.5 |
| | *Difficulty - Medium* | 187 | 18.7 |
| | *Difficulty - Hard* | 598 | 59.8 |
| | *Total* | *1000* | *100.0* |
| **Puzzle Solving** | Mate | 308 | 30.6 |
| | Cruising | 405 | 40.2 |
| | Advantage | 275 | 27.3 |
| | Others | 20 | 2.0 |
| | *Total* | *1008* | *100.0* |

moves). Among them, parsing err% + illegal mv% + forbidden% + legal mv% should equal 100%, indicating the proportion of these behaviors exhibited by the model across all attempts. We can observe that many models(e.g., Rank2: O3, Rank4: Gemini-2.5-Pro, Rank 6: Doubao-Seed-1-6-Thinking, Rank16: Doubao-1-5-Thinking-Pro and Rank 18: DeepSeek-R1) exhibit a high parsing err% rate, indicating a failure to output the specified format. This is particularly notable in some thinking models, although they often correct this in subsequent attempts and output moves in the correct format. Additionally, it can be seen that under the setting where legal moves are not provided, the illegal mv% of the models increases significantly. In Bullet mode, almost no models violate the rule prohibiting the output of thoughts, as specifically reflected in the forbidden%. The forbidden% for all models is less than 5%. As shown in Figure 6, we can observe the distribution of termination scenarios across all competitions. Over 56% of the games ended with a decisive outcome, while the remaining resulted in a draw. Among the decisive games, 31.1% were due to the model being unable to give a legal move, and 25.2% ended by checkmate. Among draws, over 30% were attributed to move limit and insufficient material, indicating that models often engage in extremely prolonged endgames and fail to conclude the match efficiently. Additionally, a small number of games ended in stalemate or due to fivefold repetition.

Tables 13, 14, and 15 present statistics on model performance in wins, losses, and draws, including metrics such as the average number of moves and the number of games won by checkmate. These data help elucidate performance differences across models. For instance, Table 13 shows that the majority of wins by the random player resulted from forfeits, which aligns with expectations, while stronger models such as O3 and Doubao-Seed-1-6-Thinking achieved a higher number of checkmate victories. As observed in Table 14, several models(e.g., Rank 9: GPT-4.1, Rank 26: Qwen3-235B-A22B) exhibit a high number of forfeit losses even when legal moves are provided, indicating potential issues with instruction adherence—specifically, the failure to output moves in the required format or to generate logically sound moves. Furthermore, the data clearly indicate that drawn games consistently involve a higher number of moves compared to decisive outcomes (wins or losses), suggesting that models often struggle to conclude games efficiently and tend to prolong them into draws. This trend is particularly pronounced among weaker models (e.g., Rank 34: Qwen3-8B, Rank 24: Random Player), which typically exhibit a higher average move count in their games compared to stronger counterparts (e.g., Rank3: Doubao-Seed-1-6-Thinking, Rank 4: Gemini-2.5-Pro, Rank 5: Qwen3-8B-Chess).

### F.2 MOVE HISTORY AFFECTION

To explore the performance difference of the model when provided with or without move history, we conducted an extra evaluation. In Blitz mode, the model's performance difference is when provided

Table 11: Whole Rating Leaderboard

| Rank | Model | Type | Legal Moves | Rating | RD | Interval | Games |
|------|-------|------|-------------|--------|-----|----------|-------|
| 1 | **Maia-1100** | - | ✗ | 2220 | 82 | (2058, 2382) | 44 |
| 2 | O3 | Standard | ✗ | 1948 | 78 | (1793, 2101) | 28 |
| 3 | Doubao-Seed-1-6-Thinking | Standard | ✓ | 1830 | 50 | (1729, 1929) | 60 |
| 4 | Gemini-2.5-Pro | Standard | ✓ | 1819 | 81 | (1659, 1979) | 18 |
| 5 | Qwen3-8B-Chess | Blitz | ✓ | 1776 | 93 | (1593, 1959) | 16 |
| 6 | Doubao-Seed-1-6-Thinking | Standard | ✗ | 1743 | 66 | (1612, 1873) | 36 |
| 7 | GPT-4.1 | Blindfold | ✓ | 1699 | 50 | (1601, 1797) | 60 |
| 8 | Doubao-Seed-1-6-Thinking | Blindfold | ✓ | 1687 | 73 | (1542, 1831) | 24 |
| 9 | GPT-4.1 | Blitz | ✓ | 1686 | 50 | (1588, 1784) | 182 |
| 10 | Claude-3-7-Sonnet | Blitz | ✓ | 1654 | 50 | (1555, 1751) | 74 |
| 11 | O3 | Blindfold | ✗ | 1636 | 115 | (1409, 1861) | 16 |
| 12 | Claude-3-7-Sonnet | Blindfold | ✓ | 1625 | 66 | (1493, 1756) | 30 |
| 13 | GPT-4.1 | Blitz | ✗ | 1623 | 50 | (1525, 1721) | 106 |
| 14 | Gemini-2.5-Pro | Standard | ✗ | 1616 | 74 | (1469, 1762) | 28 |
| 15 | Seed-Coder-8B-Chess | Blitz | ✓ | 1614 | 63 | (1490,1738) | 30 |
| 16 | Qwen3-8B-SFT | Blitz | ✓ | 1612 | 56 | (1501, 1721) | 40 |
| 17 | Doubao-1-5-Thinking-Pro | Standard | ✓ | 1598 | 63 | (1473, 1723) | 32 |
| 18 | Claude-3-7-Sonnet | Blindfold | ✗ | 1588 | 72 | (1445, 1729) | 28 |
| 19 | DeepSeek-R1 | Standard | ✓ | 1587 | 50 | (1487, 1686) | 54 |
| 20 | GPT-4.1 | Bullet | ✓ | 1583 | 50 | (1485, 1681) | 54 |
| 21 | GPT-4o | Bullet | ✓ | 1568 | 80 | (1409, 1725) | 28 |
| 22 | DeepSeek-V3 | Blitz | ✓ | 1553 | 50 | (1454, 1650) | 174 |
| 23 | Doubao-1-5-Pro | Blitz | ✓ | 1539 | 58 | (1423, 1654) | 42 |
| 24 | **Random Player** | - | ✓ | 1524 | 50 | (1425, 1621) | 284 |
| 25 | Doubao-1-5-Lite | Blitz | ✓ | 1509 | 78 | (1354, 1662) | 28 |
| 26 | Qwen3-235B-A22B | Blitz | ✓ | 1483 | 50 | (1385, 1581) | 146 |
| 27 | DeepSeek-V3 | Blitz | ✗ | 1482 | 58 | (1367, 1597) | 48 |
| 28 | Qwen3-8B-Chess | Blitz | ✗ | 1472 | 88 | (1297, 1645) | 16 |
| 29 | Claude-3-7-Sonnet | Bullet | ✓ | 1452 | 59 | (1334, 1569) | 34 |
| 30 | DeepSeek-V3 | Blindfold | ✓ | 1437 | 75 | (1290, 1584) | 24 |
| 31 | GPT-4o | Blindfold | ✓ | 1402 | 81 | (1241, 1561) | 20 |
| 32 | DeepSeek-V3 | Bullet | ✓ | 1382 | 80 | (1224, 1540) | 22 |
| 33 | Qwen3-235B-A22B | Bullet | ✓ | 1369 | 54 | (1261, 1476) | 46 |
| 34 | Qwen3-8B | Blitz | ✓ | 1335 | 65 | (1205, 1463) | 32 |
| 35 | Doubao-Seed-1-6-Thinking | Blindfold | ✗ | 1276 | 90 | (1097, 1453) | 24 |
| 36 | GPT-4.1 | Blindfold | ✗ | 1237 | 160 | (922, 1550) | 8 |
| 37 | Seed-Coder-8B-Instruct | Blitz | ✓ | 1009 | 106 | (800,1218) | 30 |

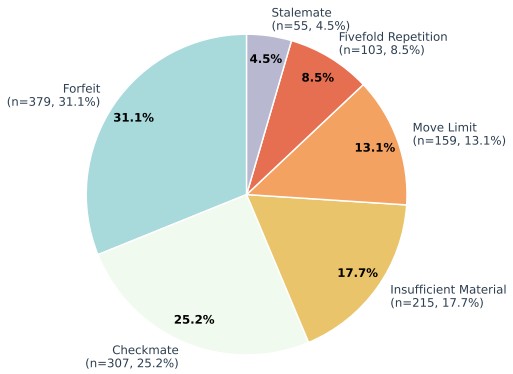

*Total Games: 1,218*

Figure 6: Distribution of Game Terminations

Table 12: ChessArena Competition Results and Performance Metrics. Mode: Play Modes(Blitz/Bullet/Standard/Blindfold); Legal: Whether legal moves were provided; Parsing Err%: Invalid format rate; Illegal Mv%: Illegal move rate; Forbidden%: Illegal thought rate in Bullet play mode; Legal Mv%: Legal move rate; Top Mv%: Top move rate. Due to the existence of draws, the sum of wins and losses does not equal the total number of games played by the model. We bold the highest top mv% among the LLMs and underline and bold the second highest.

| Rank | Model | Mode | Legal | Parsing Err% | Illegal Mv% | Forbidden% | Legal Mv% | Top Mv% |
|------|-------|------|-------|-------------|-------------|------------|-----------|---------|
| 1 | **Maia-1100** | - | × | 0.0 | 0.0 | 0.0 | 100.0 | 87.5 |
| 2 | O3 | Standard | × | 51.1 | 1.6 | 0.0 | 47.3 | **78.6** |
| 3 | Doubao-Seed-1-6-Thinking | Standard | ✓ | 2.1 | 0.3 | 0.0 | 97.6 | 51.4 |
| 4 | Gemini-2.5-Pro | Standard | ✓ | 31.8 | 0.5 | 0.0 | 67.7 | 61.8 |
| 5 | Qwen3-8B-Chess | Blitz | ✓ | 0.2 | 0.2 | 0.0 | 99.6 | 44.3 |
| 6 | Doubao-Seed-1-6-Thinking | Standard | × | 16.9 | 8.7 | 0.0 | 74.4 | 51.4 |
| 7 | GPT-4.1 | Blindfold | ✓ | 0.7 | 3.7 | 0.0 | 95.6 | 51.4 |
| 8 | Doubao-Seed-1-6-Thinking | Blindfold | ✓ | 1.0 | 1.3 | 0.0 | 97.7 | 55.1 |
| 9 | GPT-4.1 | Blitz | ✓ | 5.0 | 1.6 | 0.0 | 93.4 | 53.4 |
| 10 | Claude-3-7-Sonnet | Blitz | ✓ | 0.3 | 1.8 | 0.0 | 97.9 | 52.0 |
| 11 | O3 | Blindfold | × | 7.4 | 2.1 | 0.0 | 90.6 | **77.2** |
| 12 | Claude-3-7-Sonnet | Blindfold | ✓ | 0.1 | 1.3 | 0.0 | 98.6 | 53.7 |
| 13 | GPT-4.1 | Blitz | × | 9.8 | 18.4 | 0.0 | 71.8 | 59.4 |
| 14 | Gemini-2.5-Pro | Standard | × | 38.5 | 9.2 | 0.0 | 52.3 | **73.6** |
| 15 | Seed-Coder-8B-Chess | Blitz | ✓ | 0.7 | 0.0 | 0.0 | 99.3 | 25.9 |
| 16 | Qwen3-8B-SFT | Blitz | ✓ | 2.2 | 2.3 | 0.0 | 95.5 | 38.7 |
| 17 | Doubao-1-5-Thinking-Pro | Standard | ✓ | 29.8 | 2.4 | 0.0 | 67.8 | 53.4 |
| 18 | Claude-3-7-Sonnet | Blindfold | × | 1.4 | 24.7 | 0.0 | 73.8 | 58.5 |
| 19 | DeepSeek-R1 | Standard | ✓ | 30.8 | 1.4 | 0.0 | 67.8 | 51.0 |
| 20 | GPT-4.1 | Bullet | ✓ | 13.5 | 1.4 | 0.0 | 85.0 | 45.0 |
| 21 | GPT-4o | Bullet | ✓ | 0.0 | 1.0 | 0.0 | 99.0 | 34.4 |
| 22 | DeepSeek-V3 | Blitz | ✓ | 0.5 | 0.5 | 0.0 | 99.0 | 45.9 |
| 23 | Doubao-1-5-Pro | Blitz | ✓ | 0.1 | 5.9 | 0.0 | 94.0 | 32.2 |
| 24 | **Random Player** | - | ✓ | 0.0 | 0.0 | 0.0 | 100.0 | 40.3 |
| 25 | Doubao-1.5-Lite | Blitz | ✓ | 23.2 | 3.3 | 0.0 | 73.5 | 33.2 |
| 26 | Qwen3-235B-A22B | Blitz | ✓ | 8.4 | 3.6 | 0.0 | 88.0 | 39.3 |
| 27 | DeepSeek-V3 | Blitz | × | 1.4 | 40.8 | 0.0 | 57.8 | 43.8 |
| 28 | Qwen3-8B-Chess | Blitz | × | 0 | 30.7 | 0.0 | 69.3 | 33.8 |
| 29 | Claude-3-7-Sonnet | Bullet | ✓ | 25.3 | 0.6 | 0.0 | 74.1 | 34.6 |
| 30 | DeepSeek-V3 | Blindfold | ✓ | 4.4 | 9.4 | 0.0 | 86.2 | 33.7 |
| 31 | GPT-4o | Blindfold | ✓ | 0.0 | 1.8 | 0.0 | 97.1 | 37.4 |
| 32 | DeepSeek-V3 | Bullet | ✓ | 0.0 | 2.5 | 3.4 | 96.9 | 32.9 |
| 33 | Qwen3-235B-A22B | Bullet | ✓ | 0.0 | 2.7 | 0.5 | 96.8 | 35.3 |
| 34 | Qwen3-8B | Blitz | ✓ | 1.6 | 1.3 | 0.0 | 97.1 | 32.9 |
| 35 | Doubao-Seed-1-6-Thinking | Blindfold | × | 2.8 | 39.2 | 0.0 | 58.0 | 54.0 |
| 36 | GPT-4.1 | Blindfold | × | 2.8 | 34.6 | 0.0 | 62.6 | 62.2 |
| 37 | Seed-Coder-8B-Instruct | Blitz | ✓ | 14.9 | 43.5 | 0.0 | 41.6 | 34.0 |

with or without move history, as can be seen from Table 18. Providing or not providing move history does not significantly affect the model's performance, with evaluation metrics fluctuating within the range of 1% to 5%. As we previously mentioned, providing move history primarily serves to allow the model to access historical information, thereby helping to avoid fivefold repetition draw. Without legal move constraints, PGN notation improves model performance, likely because PGN notation aligns better with the model's training corpus. The model is more familiar with this method of representing move history. In our ChessArena, as long as both sides are provided with consistent information, it does not introduce unfairness that might arise from differences in move history representation.

# G  ANALYSIS

## G.1  WHY DO LLMS FAIL IN CHESS?

The results presented above indicate that LLMs exhibit relatively poor performance in chess. We attribute this deficiency primarily to the following factors:

**Lack of Instruction-Following Capability**   In Table 12, the metrics Parsing Err%, Illegal Mv%, and Forbidden% serve as indicators of instruction adherence; higher values denote a greater frequency of errors. Specifically, these failures manifest in three ways:

Table 13: ChessArena Competition wining games statistics: Wins: Number of games won; Winning Move: Average move of wining games; Checkmate / Forfeit: Number of games won by checkmate / forfeit

| Rank | Model | Mode | Legal | Wins | Winning Move | Checkmate | Forfeit |
|------|-------|------|-------|------|--------------|-----------|---------|
| 1 | **Maia-1100** | - | ✗ | 44 | 21 | 40 | 4 |
| 2 | O3 | Standard | ✗ | 15 | 18 | 7 | 8 |
| 3 | Doubao-Seed-1-6-Thinking | Standard | ✓ | 32 | 36 | 20 | 12 |
| 4 | Gemini-2.5-Pro | Standard | ✓ | 10 | 26 | 10 | 0 |
| 5 | Qwen3-8B-Chess | Blitz | ✓ | 7 | 46 | 7 | 0 |
| 6 | Doubao-Seed-1-6-Thinking | Standard | ✗ | 15 | 38 | 9 | 6 |
| 7 | GPT-4.1 | Blindfold | ✓ | 17 | 23 | 8 | 9 |
| 8 | Doubao-Seed-1-6-Thinking | Blindfold | ✓ | 5 | 24 | 5 | 0 |
| 9 | GPT-4.1 | Blitz | ✓ | 54 | 29 | 34 | 20 |
| 10 | Claude-3-7-Sonnet | Blitz | ✓ | 13 | 29 | 5 | 8 |
| 11 | O3 | Blindfold | ✗ | 16 | 19 | 1 | 15 |
| 12 | Claude-3-7-Sonnet | Blindfold | ✓ | 4 | 35 | 4 | 0 |
| 13 | GPT-4.1 | Blitz | ✗ | 67 | 14 | 14 | 53 |
| 14 | Gemini-2.5-Pro | Standard | ✗ | 16 | 22 | 2 | 14 |
| 15 | Seed-Coder-8B-Chess | Blitz | ✓ | 9 | 41 | 9 | 0 |
| 16 | Qwen3-8B-SFT | Blitz | ✓ | 10 | 63 | 10 | 0 |
| 17 | Doubao-1-5-Thinking-Pro | Standard | ✓ | 4 | 40 | 3 | 1 |
| 18 | Claude-3-7-Sonnet | Blindfold | ✗ | 9 | 20 | 1 | 8 |
| 19 | DeepSeek-R1 | Standard | ✓ | 9 | 42 | 8 | 1 |
| 20 | GPT-4.1 | Bullet | ✓ | 7 | 12 | 3 | 4 |
| 21 | GPT-4o | Bullet | ✓ | 2 | 17 | 2 | 0 |
| 22 | DeepSeek-V3 | Blitz | ✓ | 38 | 32 | 12 | 26 |
| 23 | Doubao-1-5-Pro | Blitz | ✓ | 10 | 53 | 9 | 1 |
| 24 | **Random Player** | - | ✓ | 91 | 44 | 4 | 87 |
| 25 | Doubao-1.5-Lite | Blitz | ✓ | 4 | 40 | 1 | 3 |
| 26 | Qwen3-235B-A22B | Blitz | ✓ | 28 | 30 | 10 | 18 |
| 27 | DeepSeek-V3 | Blitz | ✗ | 34 | 8 | 0 | 34 |
| 28 | Qwen3-8B-Chess | Blitz | ✗ | 7 | 15 | 0 | 7 |
| 29 | Claude-3-7-Sonnet | Bullet | ✓ | 0 | / | 0 | 0 |
| 30 | DeepSeek-V3 | Blindfold | ✓ | 6 | 19 | 4 | 2 |
| 31 | GPT-4o | Blindfold | ✓ | 4 | 22 | 3 | 1 |
| 32 | DeepSeek-V3 | Bullet | ✓ | 1 | 34 | 1 | 0 |
| 33 | Qwen3-235B-A22B | Bullet | ✓ | 6 | 46 | 1 | 5 |
| 34 | Qwen3-8B | Blitz | ✓ | 5 | 31 | 0 | 5 |
| 35 | Doubao-Seed-1-6-Thinking | Blindfold | ✗ | 1 | 30 | 0 | 1 |
| 36 | GPT-4.1 | Blindfold | ✗ | 0 | / | 0 | 0 |
| 37 | Seed-Coder-8B-Instruct | Blitz | ✓ | 0 | / | 0 | 0 |

- Parsing Errors: The model fails to adhere to the specified output format, rendering the move unextractable.
- Illegal Moves: The model fails to select a legal move, even when the list of Legal Moves is explicitly provided in the prompt. This is particularly evident in weaker models (e.g., Rank 23: Doubao-1-5-Pro, Rank 26: Qwen3-235B-A22B).
- Forbidden Tokens: The model generates "thinking tokens" during Bullet play mode (where speed is critical). While rare in the ChessArena benchmark, this was observed in Rank 32: DeepSeek-V3 (3.4%).

Elevated error rates in these metrics increase the likelihood of the model failing to produce a legal move after multiple retries, resulting in a forfeit.

**Deficiency in Strategic Reasoning**   Most models fail to infer valid moves solely from the board state (FEN). As shown in Table 2, only "Thinking" models (e.g., DeepSeek-R1, Doubao-Seed-1-6-Thinking, O3, Gemini-2.5-Pro) achieve a Precision% exceeding 95% when identifying legal moves for specific pieces. However, actual gameplay requires the model to validate moves for all pieces globally, meaning identification errors accumulate. Consequently, we observe that only models achieving >90% in both Precision% and Recall% on the Basic Understanding task can play effectively without an explicitly provided list of legal moves (see Table 3).

Table 14: ChessArena Competition Losing games statistics: Losses: Number of games lost; Losses Move: Average move of lost games; Checkmate / Forfeit: Number of games lost by checkmate / forfeit

| Rank | Model | Mode | Legal | Losses | Losses Move | Checkmate | Forfeit |
|---|---|---|---|---|---|---|---|
| 1 | **Maia-1100** | - | ✗ | 0 | / | 0 | 0 |
| 2 | O3 | Standard | ✗ | 13 | 20 | 7 | 6 |
| 3 | Doubao-Seed-1-6-Thinking | Standard | ✓ | 7 | 24 | 7 | 0 |
| 4 | Gemini-2.5-Pro | Standard | ✓ | 2 | 48 | 2 | 0 |
| 5 | Qwen3-8B-Chess | Blitz | ✓ | 0 | / | 0 | 0 |
| 6 | Doubao-Seed-1-6-Thinking | Standard | ✗ | 17 | 21 | 8 | 9 |
| 7 | GPT-4.1 | Blindfold | ✓ | 12 | 35 | 9 | 3 |
| 8 | Doubao-Seed-1-6-Thinking | Blindfold | ✓ | 1 | 40 | 0 | 1 |
| 9 | GPT-4.1 | Blitz | ✓ | 44 | 30 | 32 | 12 |
| 10 | Claude-3-7-Sonnet | Blitz | ✓ | 18 | 29 | 16 | 2 |
| 11 | O3 | Blindfold | ✗ | 0 | / | 0 | 0 |
| 12 | Claude-3-7-Sonnet | Blindfold | ✓ | 6 | 34 | 6 | 0 |
| 13 | GPT-4.1 | Blitz | ✗ | 38 | 20 | 4 | 34 |
| 14 | Gemini-2.5-Pro | Standard | ✗ | 12 | 16 | 1 | 11 |
| 15 | Seed-Coder-8B-Chess | Blitz | ✓ | 3 | 37 | 1 | 2 |
| 16 | Qwen3-8B-SFT | Blitz | ✓ | 7 | 36 | 2 | 5 |
| 17 | Doubao-1-5-Thinking-Pro | Standard | ✓ | 2 | 20 | 0 | 2 |
| 18 | Claude-3-7-Sonnet | Blindfold | ✗ | 17 | 28 | 3 | 14 |
| 19 | DeepSeek-R1 | Standard | ✓ | 6 | 22 | 6 | 0 |
| 20 | GPT-4.1 | Bullet | ✓ | 6 | 34 | 6 | 0 |
| 21 | GPT-4o | Bullet | ✓ | 9 | 21 | 9 | 0 |
| 22 | DeepSeek-V3 | Blitz | ✓ | 43 | 48 | 38 | 5 |
| 23 | Doubao-1-5-Pro | Blitz | ✓ | 6 | 19 | 4 | 2 |
| 24 | **Random Player** | - | ✓ | 67 | 47 | 67 | 0 |
| 25 | Doubao-1.5-Lite | Blitz | ✓ | 11 | 36 | 4 | 7 |
| 26 | Qwen3-235B-A22B | Blitz | ✓ | 45 | 31 | 27 | 18 |
| 27 | DeepSeek-V3 | Blitz | ✗ | 14 | 11 | 0 | 14 |
| 28 | Qwen3-8B-Chess | Blitz | ✗ | 9 | 6 | 0 | 9 |
| 29 | Claude-3-7-Sonnet | Bullet | ✓ | 6 | 24 | 6 | 0 |
| 30 | DeepSeek-V3 | Blindfold | ✓ | 6 | 51 | 6 | 0 |
| 31 | GPT-4o | Blindfold | ✓ | 1 | 8 | 1 | 0 |
| 32 | DeepSeek-V3 | Bullet | ✓ | 5 | 44 | 5 | 0 |
| 33 | Qwen3-235B-A22B | Bullet | ✓ | 13 | 39 | 13 | 0 |
| 34 | Qwen3-8B | Blitz | ✓ | 15 | 48 | 15 | 0 |
| 35 | Doubao-Seed-1-6-Thinking | Blindfold | ✗ | 23 | 15 | 23 | 0 |
| 36 | GPT-4.1 | Blindfold | ✗ | 8 | 25 | 8 | 0 |
| 37 | Seed-Coder-8B-Instruct | Blitz | ✓ | 31 | 4 | 3 | 28 |

Furthermore, even when models produce valid moves, the quality remains suboptimal. While models generally outperform random players in TR% and MAR% (metrics measuring move quality in Table 3), they fall significantly short of human baselines (i.e., Maia-1100).

Weaker models frequently struggle to convert advantages into checkmates, leading to unnecessary draws. In advantageous positions, instead of executing a decisive sequence, these models often select erratic moves that force the game into a draw via move limits or insufficient material (as evidenced by the distribution in Figure 6). Figure 7 illustrates a specific instance where DeepSeek-R1 fails to identify a simple one-move checkmate, choosing a mediocre move instead.

Fundamentally, these failures point to a deficit in strategic reasoning capabilities. Our tasks are analogous to propositional logic problems: the model must derive a solution based on known conditions (FEN, Position, or Legal Moves) and established knowledge (game rules). The limited reasoning ability demonstrated by models in the ChessArena environment highlights a critical area requiring further research and optimization.

Table 15: ChessArena Competition Drawing games statistics: Draws: Number of games drawn; Draws Move: Average move of drawn games; Stalemate / Move Limit / Insufficient Material / Fivefold Repetition: Number of games drawn by stalemate / move limit / insufficient material / fivefold repetition

| Rank | Model | Mode | Legal | Draws | Draws Move | Stalemate | Move Limit | Insufficient Material | Fivefold Repetition |
|------|-------|------|-------|-------|-----------|-----------|-----------|----------------------|---------------------|
| 1 | **Maia-1100** | - | × | 0 | / | 0 | 0 | 0 | 0 |
| 2 | O3 | Standard | × | 0 | / | 0 | 0 | 0 | 0 |
| 3 | Doubao-Seed-1-6-Thinking | Standard | ✓ | 21 | 62 | 6 | 0 | 15 | 0 |
| 4 | Gemini-2.5-Pro | Standard | ✓ | 6 | 68 | 1 | 0 | 5 | 0 |
| 5 | Qwen3-8B-Chess | Blitz | ✓ | 9 | 63 | 2 | 1 | 6 | 0 |
| 6 | Doubao-Seed-1-6-Thinking | Standard | × | 4 | 66 | 0 | 0 | 4 | 0 |
| 7 | GPT-4.1 | Blindfold | ✓ | 31 | 70 | 1 | 1 | 11 | 18 |
| 8 | Doubao-Seed-1-6-Thinking | Blindfold | ✓ | 18 | 73 | 0 | 2 | 8 | 8 |
| 9 | GPT-4.1 | Blitz | ✓ | 84 | 74 | 10 | 7 | 51 | 16 |
| 10 | Claude-3-7-Sonnet | Blitz | ✓ | 43 | 71 | 2 | 8 | 30 | 3 |
| 11 | O3 | Blindfold | × | 0 | / | 0 | 0 | 0 | 0 |
| 12 | Claude-3-7-Sonnet | Blindfold | ✓ | 19 | 73 | 2 | 3 | 8 | 6 |
| 13 | GPT-4.1 | Blitz | × | 1 | 100 | 0 | 1 | 0 | 0 |
| 14 | Gemini-2.5-Pro | Standard | × | 0 | / | 0 | 0 | 0 | 0 |
| 15 | Seed-Coder-8B-Chess | Blitz | ✓ | 19 | 60 | 11 | 1 | 1 | 6 |
| 16 | Qwen3-8B-SFT | Blitz | ✓ | 23 | 90 | 0 | 2 | 21 | 0 |
| 17 | Doubao-1-5-Thinking-Pro | Standard | ✓ | 26 | 69 | 0 | 2 | 24 | 0 |
| 18 | Claude-3-7-Sonnet | Blindfold | × | 2 | 61 | 0 | 0 | 1 | 1 |
| 19 | DeepSeek-R1 | Standard | ✓ | 39 | 72 | 3 | 5 | 28 | 3 |
| 20 | GPT-4.1 | Bullet | ✓ | 41 | 75 | 2 | 4 | 20 | 15 |
| 21 | GPT-4o | Bullet | ✓ | 17 | 82 | 0 | 7 | 1 | 9 |
| 22 | DeepSeek-V3 | Blitz | ✓ | 92 | 87 | 19 | 29 | 29 | 15 |
| 23 | Doubao-1-5-Pro | Blitz | ✓ | 26 | 102 | 2 | 13 | 8 | 3 |
| 24 | **Random Player** | - | ✓ | 126 | 110 | 11 | 90 | 23 | 2 |
| 25 | Doubao-1.5-Lite | Blitz | ✓ | 13 | 91 | 1 | 2 | 5 | 5 |
| 26 | Qwen3-235B-A22B | Blitz | ✓ | 74 | 94 | 5 | 19 | 46 | 4 |
| 27 | DeepSeek-V3 | Blitz | × | 0 | / | 0 | 0 | 0 | 0 |
| 28 | Qwen3-8B-Chess | Blitz | × | 0 | / | 0 | 0 | 0 | 0 |
| 29 | Claude-3-7-Sonnet | Bullet | ✓ | 28 | 92 | 0 | 0 | 28 | 0 |
| 30 | DeepSeek-V3 | Blindfold | ✓ | 12 | 86 | 0 | 0 | 12 | 0 |
| 31 | GPT-4o | Blindfold | ✓ | 15 | 65 | 0 | 0 | 15 | 0 |
| 32 | DeepSeek-V3 | Bullet | ✓ | 16 | 81 | 0 | 0 | 16 | 0 |
| 33 | Qwen3-235B-A22B | Bullet | ✓ | 27 | 73 | 0 | 0 | 27 | 0 |
| 34 | Qwen3-8B | Blitz | ✓ | 12 | 126 | 0 | 0 | 12 | 0 |
| 35 | Doubao-Seed-1-6-Thinking | Blindfold | × | 0 | / | 0 | 0 | 0 | 0 |
| 36 | GPT-4.1 | Blindfold | × | 0 | / | 0 | 0 | 0 | 0 |
| 37 | Seed-Coder-8B-Instruct | Blitz | ✓ | 0 | / | 0 | 0 | 0 | 0 |

Table 16: Average Conversation Turn Count in Blindfold move selection (Without Legal Move Provision)

| Model | Thinking | Successful Turn Count | Failed Turn Count |
|-------|----------|----------------------|-------------------|
| GPT-4.1 | × | 94 | 94 |
| DeepSeek-V3 | × | 94 | 88 |
| Qwen3-235B-A22B | × | 96 | 86 |
| Claude-3.7-Sonnet | × | 92 | 94 |
| DeepSeek-R1 | ✓ | 72 | 112 |
| Doubao-Seed-1.6-Thinking | ✓ | 70 | 112 |
| Gemini-2.5-Pro | ✓ | 88 | 105 |
| O3 | ✓ | 88 | 129 |

Table 17: Puzzle Solving Accuracy: Blitz/Standard Prompt Template and Legal Moves not Provided

| Model or Engine | Puzzle Solving Accuracy (%) | | | | | | | |
|-----------------|---------|----------|-----------|-----------|-----------|-----------|-----------|---------|
| | 200-600 | 600-1000 | 1000-1400 | 1400-1800 | 1800-2200 | 2200-2600 | 2600-3000 | Overall |
| GPT-4.1 | 44.1 | 29.4 | 18.2 | 12.6 | 4.2 | 2.8 | 0.0 | **15.9** |
| Claude-3-7-Sonnet | 18.2 | 16.1 | 4.9 | 4.2 | 5.6 | 1.4 | 0.0 | 7.2 |
| DeepSeek-V3 | 2.1 | 2.1 | 2.1 | 2.8 | 0.7 | 1.4 | 0.0 | 1.6 |
| DeepSeek-V3.1(Non-thinking) | 9.8 | 7.7 | 4.9 | 2.1 | 2.8 | 1.4 | 0.7 | 4.2 |
| Qwen3-235B-A22B(Non-thinking) | 16.7 | 12.5 | 7.2 | 4.5 | 5.0 | 4.2 | 0.0 | 7.1 |
| Qwen3-8B | 0.0 | 0.0 | 0.0 | 0.0 | 0.0 | 0.0 | 0.0 | 0.0 |
| Qwen3-8B-Chess(SFT+RL) | 7.0 | 8.4 | 4.2 | 2.8 | 0.7 | 0.7 | 0.0 | 3.4 |
| O3 | 95.8 | 93.0 | 87.4 | 68.5 | 44.8 | 16.8 | 4.2 | **58.6** |
| Gemini-2.5-Pro | 45.2 | 39.2 | 19.6 | 14.0 | 2.8 | 3.5 | 0.7 | 19.9 |
| Doubao-Seed-1-6-Thinking | 21.0 | 22.4 | 13.3 | 4.9 | 4.9 | 0.7 | 0.0 | 9.6 |
| DeepSeek-R1 | 18.9 | 13.3 | 11.2 | 1.9 | 2.1 | 1.4 | 0.7 | 7.1 |

Table 18: GPT-4.1 Performance in Blitz Mode: Impact of Move History

| Evaluation Condition | LR (%) | TR (%) |
|---|---|---|
| *With Legal Move Constraints* | | |
| Without Move History | 98.2 | 25.0 |
| With Move History(List of UCI) | 97.2 | 29.0 |
| With Move History(PGN) | 97.0 | 28.7 |
| *Without Legal Move Constraints* | | |
| Without Move History | 68.4 | 28.4 |
| With Move History(List of UCI) | 70.2 | 26.2 |
| With Move History(PGN) | 75.4 | 34.5 |

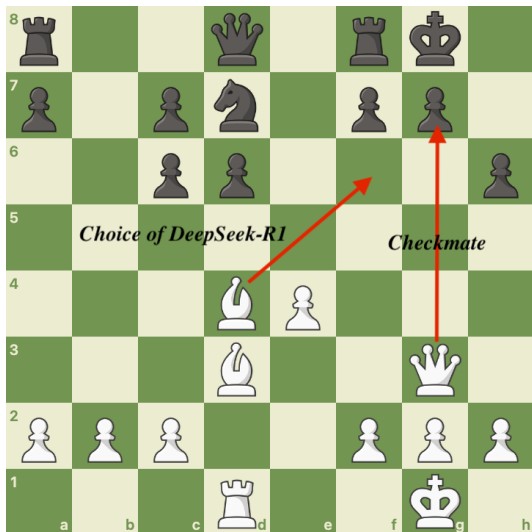

Figure 7: DeepSeek-R1 fails to checkmate. Left: DeepSeek-R1's choice; Right: The optimal Checkmate Move.

## G.2 BLINDFOLD ANALYSIS

We discussed the different behaviour between thinking and non-thinking LLMs in Blindfold chess games. Non-thinking models frequently exhibit laziness and shortcut-taking behavior, while thinking models genuinely attempt to reconstruct the board, but this task proves excessively difficult. Table 19 is a case study of different models' responses in a Blindfold chess game. Thinking models such as Doubao-Seed-1-6-thinking always try to reconstruct the chessboard state, but it's too difficult. GPT-4.1 often takes shortcuts by directly following the last move made on its own side in the conversation history. Claude-3-7-Sonnet does not explicitly reconstruct the chessboard state but still retains some ability to analyze the game situation. Overall, chess gameplay under blindfold play modes proves excessively challenging, making it difficult for models to reconstruct the chessboard state and analyze potential moves accurately. Non-thinking models demonstrate virtually no capability for genuine board reconstruction. Among thinking models, only the O3 model exhibits satisfactory performance. This highlights substantial room for improvement in models' multi-turn conversation memorization capabilities.

Table 19: A Case Study of Blindfold chess game of different LLMs

**Query Details:**
System prompt is omitted.
**Conversation History (Recent Moves):**
80+ Conversation Turns are omitted.
- **user**: The opponent's last move is b7c7.
- **assistant**: My move: e3e7
- **user**: The opponent's last move is c7c6.
- **assistant**: My move: e7c7
- **user**: The opponent's last move is c6c7.
- **assistant**: My move: e8e7
- **user**: The opponent's last move is c7b6.
- **assistant**: My move: **f1h3**
- **user (Final Query)**: The opponent's last move is b6a6. What is the best Move? Think it step by step.

**Case 1: Doubao-Seed-1-6-thinking Failed to Reconstruct the Chessboard: It has fallen into the repeater problem.**

Got it, let's try to figure out the current position first. I need to recap the move history to get the pieces on the board.
First, I'm White, since the first move was g1f3 (White's knight). Let's list the key moves for both sides, focusing on piece locations:
**White's moves (me):** - Knights: Started on g1 and b1. g1f3 -> f3g5 -> g5f7 -> f7h8 -> h8f7 ->
f7h8 -> h8f7 -> f7h8 -> h8f7 -> f7h8 -> h8f7 -> f7h8 -> h8f7 -> f7h8 -> h8f7 ->
f7h8 -> h8f7 -> f7h8 -> h8f7 -> f7h8 -> h8f7 -> f7h8 -> h8f7 -> f7h8 -> h8f7 ->
f7h8 -> h8f7 -> f7h8 -> h8f7 -> f7h8 -> h8f7 -> f7h8 -> h8f7 -> f7h8 -> h8f7 ->
f7h8 -> h8f7 -> f7h8 -> h8f7 -> f7h8 -> h8f7 -> f7h8 -> h8f7 -> f7h8 -> h8f7 ->
f7h8 -> h8f7 -> f7h8 -> h8f7 -> f7h8 -> h8f7 -> f7h8 -> h8f7 -> f7h8 -> h8f7 ->
f7h8 -> h8f7 -> f7h8 -> h8f7 -> f7h8 -> h8f7 -> f7h8 -> h8f7 -> f7h8 -> h8f7 ->
f7h8 -> h8f7 -> f7h8 -> h8f7 -> f7h8 -> h8f7 -> f7h8 -> h8f7 -> f7h8 -> h8f7 ->
f7h8 -> h8f7 -> f7h8 -> h8f7 -> f7h8 -> h8f7 -> f7h8 -> h8f7 -> f7h8 -> h8f7 ->
f7h8 -> h8f7 -> f7h8 -> h8f7 -> f7h8 -> h8f7 -> f7h8 -> h8f7 -> f7h8 -> h8f7 ->
f7h8 -> h8f7 -> f7h8 -> h8f7 -> f7h8 -> h8f7 -> f7h8 -> h8f7... **repetition...** words are omited.

**Case 2: GPT-4.1 takes a shortcut: It continues moving from the landing point(i.e., h3) of the last piece in the Query.**

Table 19 (Continued): Blindfold Model Performance

My move:
```
h3e6
```

**Case 3: Claude-3-7-Sonnet analyzes the chessboard and tries to give a better move.**

I'll analyze the position carefully to find the best move.
**Step 1: Confirm Current Material and Position** - White's pieces: Based on moves, we have a bishop on h3 (f1h3), a rook/queen on e7 (e8e7, previously e3e7 -> e7c7 -> e8e7), and other pieces (knights, remaining pawns) whose positions need to be inferred. - Black's pieces: Only the king is confirmed on a6 (b6a6), and other pieces (like pawns on a6's original position) have been captured or moved. - Material advantage: White has at least a bishop + rook/queen vs. Black's lone king—clear advantage, goal is to checkmate.
**Step 2: Evaluate Candidate Moves** 1. **e7a7**: Directly attacks the Black king on a6. If e7 is a queen/rook, e7a7 would be check. But need to confirm if a7 is occupied—no Black pieces are known to be on a7, so this could be checkmate. 2. **h3c8**: Moves the bishop to c8, controlling b7 (a square the Black king might flee to if attacked). This prepares for a subsequent checkmate but is not immediate. 3. **e7e6**: Moves the e7 piece to e6, controlling d5 and f5—limits the king's potential escape to the center, but is slower than direct attack. 4. **b4b5**: If White has a pawn on b4, b4b5 would check the king (attacking a6), but there's no record of a b4 pawn in the move history, so this is risky.
**Step 3: Select the Best Move** e7a7 is the most efficient: it directly targets the Black king, and since there are no known Black pieces blocking a7, this move will put the king in check. If the king has no legal moves to escape (a6 can only move to a5, b5, or b6—b6 is empty but controlled by White's pieces), this becomes checkmate.
My move:
```
e7a7
```

## G.3 THE GENERALIZATION OF CHESS REASONING TRAINING

In Table 20, we present the performance of Qwen3-8B and its training versions on other important benchmarks, including LiveCodeBench-v5 (Jain et al., 2024), AIME2025 (MAA, 2025), ZebraLogic (Lin et al., 2025), BigCodeBench (Zhuo et al.), CruxEval (Gu et al.) and DROP (Dua et al., 2019)). They are well-known benchmarks in code generation, logical reasoning, and mathematics domains. These models were evaluated under non-thinking (cause our model is trained under a non-thinking chat template) conditions with a 4k context length, temperature of 0.6, and top-p of 0.95. The results demonstrate that our chess domain-trained Qwen3-8B model maintains comparable performance across other benchmarks without significant score degradation. Notably, we observe performance improvements on AIME2025 and ZebraLogic, indicating that chess domain-specific training exhibits positive transfer effects to other domains. Furthermore, the cross-domain performance improvements appear to stem primarily from RL, demonstrating the effectiveness of chess RL training. We observed substantial improvements in the model's performance on Zebralogic tasks ($25.9 \rightarrow 48$), with score increases across all puzzle categories: small puzzles, medium puzzles, large puzzles, and XL puzzles ($70.3/12.1/0.0/0.0 \rightarrow 92.2/55.4/14.5/0.5$). Models trained on chess demonstrate significant performance gains on puzzle-solving and related reasoning tasks. We analyzed the model's responses for a case study to investigate the underlying factors contributing to the performance improvements, as shown in Table 21. After chess training, the model became significantly more systematic and logically clear in solving such problems, with a final verification process that markedly improved the model's success rate in addressing these problems. Based on Puzzle 2 and Puzzle 3, we can observe that the model, after being trained on chess data, indeed

exhibits an increased number of reasoning steps and a more pronounced process of self-verification. This enhancement significantly strengthens the model's ability to solve large and medium-sized puzzles.

To better evaluate the generalization capabilities of models after Chess post-training, we designed a series of ablation studies. These experiments included various mixed settings such as training only on Math, only on Code, Math + Code, and Math + Code + Chess. The Math and Code data utilized were sourced from Eurus-2-RL-data[5]. The data split quantities used are as follows: Math: 10k, Code: 10k, Math + Code: 5k + 5k, Math + Code + Chess: 3.3k + 3.3k + 3.3k, and Math + Chess: 8k + 2k.

The detailed experimental results are presented in Table 20.

1. Models fine-tuned via Chess Supervised Fine-Tuning (SFT), regardless of whether they subsequently undergo Code, Math, or Chess Reinforcement Learning (RL), consistently demonstrate a degree of generalization ability to the logical reasoning benchmark, ZebraLogic. Specifically, models that underwent Chess SFT show a significant improvement in their ZebraLogic scores after the RL phase.

2. Incorporating a certain amount of Chess data into the RL dataset contributes to the generalization of Code capabilities, particularly on the LiveCodeBench. Comparisons (e.g., Math + Chess - RL / Math + Code + Chess - RL versus Math - RL / Math + Code - RL) reveal that RL models augmented with Chess data achieve a improvement in their LiveCodeBench scores.

3. However, the use of Chess SFT may lead to performance degradation on other benchmarks (e.g., BigCode-Bench, CruxEval); mixing Chess data for RL could also result in a slight performance decline on mathematical tasks (i.e., AIME2025).

Overall, training on Chess exhibits tangible generalization capabilities across different domains, suggesting significant avenues for future research.

Table 20: Performance of Qwen3-8B and its Post-Trained Variants on External Benchmarks

| Model Variant | LiveCodeBench | AIME2025 | ZebraLogic | BigCodeBench | CruxEval | DROP |
|---|---|---|---|---|---|---|
| **Qwen3-8B (Baseline)** | 25.19 | 18.61 | 25.90 | 41.32 | 73.25 | 85.15 |
| **I. Chess Post-train** | | | | | | |
| Qwen3-8B-Chess-SFT-Stage2 | 27.48 | 15.43 | 30.40 | 41.40 | 68.00 | 82.83 |
| Qwen3-8B-Chess (SFT+RL) | 25.19 | 19.30 | 48.00 | 39.82 | 72.25 | 83.58 |
| **II. RL with Chess-SFT** | | | | | | |
| +Math-RL | 25.19 | 19.70 | 42.40 | 39.39 | 73.25 | 84.12 |
| +Math+Chess-RL | 26.72 | 19.32 | 39.00 | 40.09 | 72.09 | 84.01 |
| +Math+Code-RL | 28.24 | 18.61 | 38.30 | 40.26 | 71.25 | 83.34 |
| +Math+Code+Chess-RL | 30.53 | 17.96 | 38.00 | 40.30 | 72.22 | 83.84 |
| **III. RL without Chess-SFT** | | | | | | |
| +Math-RL | 25.19 | 21.30 | 28.00 | 42.37 | 77.47 | 85.53 |

---

[5]https://huggingface.co/datasets/PRIME-RL/Eurus-2-RL-Data

Table 21: ZebraLogic Case Studies

**Puzzle 1: Qwen3-8B-Chess performs better due to its more rigorous reasoning steps.**

**Puzzle to Solve**:
There are 4 houses, numbered 1 to 4 from left to right, as seen from across the street. Each house is occupied by a different person. Each house has a unique attribute for each of the following characteristics:
- Each person has a unique name: 'Alice', 'Eric', 'Arnold', 'Peter'
- Each person has an occupation: 'artist', 'engineer', 'teacher', 'doctor'
- People have unique favorite book genres: 'fantasy', 'science fiction', 'mystery', 'romance'
- People use unique phone models: 'google pixel 6', 'iphone 13', 'oneplus 9', 'samsung galaxy s21'
**Clues:** 1. The person who is an engineer is directly left of the person who uses a Samsung Galaxy S21.
2. The person who loves fantasy books is in the second house.
3. Alice is not in the second house.
4. Eric is the person who is a teacher.
5. The person who uses a Samsung Galaxy S21 is the person who loves fantasy books.
6. The person who uses an iPhone 13 is the person who loves science fiction books.
7. The person who loves science fiction books is somewhere to the left of the person who uses a OnePlus 9.
8. The person who uses a OnePlus 9 is Arnold.
9. The person who is a doctor is the person who loves mystery books.
10. The person who uses an iPhone 13 is the person who is a teacher.

**Ground-Truth**

```
{
    "House 1": { "Name": "Alice", "Occupation": "engineer", "BookGenre": "romance",
"PhoneModel": "google pixel 6" },
    "House 2": { "Name": "Peter", "Occupation": "artist", "BookGenre": "fantasy",
"PhoneModel": "samsung galaxy s21" },
    "House 3": { "Name": "Eric", "Occupation": "teacher", "BookGenre": "science fiction",
"PhoneModel": "iphone 13" },
    "House 4": { "Name": "Arnold", "Occupation": "doctor", "BookGenre": "mystery",
"PhoneModel": "oneplus 9" }
}
```

Table 21 (Continued): ZebraLogic Case Study

**Case 1: Response from Qwen3-8B. The reasoning process lacks clarity and violates clue 3,9**

**Reasoning**:
Starting with Clue 2, we know the person who loves fantasy books is in House 2. Clue 5 says the person who uses a Samsung Galaxy S21 is the one who loves fantasy books, so House 2 uses the Samsung Galaxy S21. Clue 1 says the engineer is directly left of the person using a Samsung Galaxy S21, so the engineer must be in House 1. Clue 10 states the person using an iPhone 13 is the teacher, and Clue 4 says Eric is the teacher, so Eric uses an iPhone 13. Clue 6 says the person using an iPhone 13 loves science fiction books, so Eric loves science fiction books. Clue 7 says the person who loves science fiction books is to the left of the person using a OnePlus 9, so Eric must be in a house to the left of the OnePlus 9 user. Clue 8 says the person using a OnePlus 9 is Arnold, so Arnold is in a house to the right of Eric. Clue 3 says Alice is not in the second house, and we already know House 2 has the person who loves fantasy books. Clue 9 says the doctor loves mystery books. Clue 10 already assigns Eric as the teacher. Clue 4 confirms Eric is the teacher. Clue 10 also says the person using an iPhone 13 is the teacher, so Eric uses an iPhone 13. Using this information, we can assign the rest of the attributes.
**Solution**:
{
    "House 1": { "Name": "Peter", "Occupation": "engineer", "BookGenre": "mystery", "PhoneModel": "google pixel 6" },
    "House 2": { "Name": "Alice", "Occupation": "doctor", "BookGenre": "fantasy", "PhoneModel": "samsung galaxy s21" },
    "House 3": { "Name": "Eric", "Occupation": "teacher", "BookGenre": "science fiction", "PhoneModel": "iphone 13" },
    "House 4": { "Name": "Arnold", "Occupation": "artist", "BookGenre": "romance", "PhoneModel": "oneplus 9" }
}

**Case 2: Response from Qwen3-8B-Chess. The entire reasoning process is divided into 6 systematic steps with clear logical structure, including a final verification process.**

Table 21 (Continued): ZebraLogic Case Study

**Reasoning:**
Let's break down the clues step by step to solve the puzzle.
**Step1: Key Information** - Names: Alice, Eric, Arnold, Peter
- Occupations: artist, engineer, teacher, doctor
- BookGenres: fantasy, science fiction, mystery, romance
- PhoneModels: google pixel 6, iphone 13, oneplus 9, samsung galaxy s21
**Step 2: Clues Analysis**
Clue 1: The person who is an engineer is directly left of the person who uses a Samsung Galaxy S21. - This means the engineer is in a house with a number one less than the house of the Samsung Galaxy S21 user.
Clue 2: The person who loves fantasy books is in the second house. - House 2's BookGenre is fantasy.
- Eric's Occupation = teacher.
<For brevity and readability, the following of this step has been omitted.>
**Step3: Step-by-Step Deduction**
From Clue 5 and 2: - House 2's PhoneModel = samsung galaxy s21 (since BookGenre = fantasy).
From Clue 1: - The engineer is directly left of the Samsung Galaxy S21 user (House 2).
- So the engineer must be in House 1 (since only House 1 is directly left of House 2).
From Clue 10 and 4: - Eric is a teacher and uses an iPhone 13.
- So Eric's PhoneModel = iphone 13, and from Clue 6, his BookGenre = science fiction.
<Omitted.>
**Step 4: Now, let's determine the positions**
House 1: - Occupation = engineer
- Name = ? (Not Eric, not Arnold, not Alice in House 2)
- Possible names: Alice or Peter
House 2: - BookGenre = fantasy
- PhoneModel = samsung galaxy s21
- Name = ? (Not Alice, not Eric)
- Possible names: Arnold or Peter
Continued on the next page.

Table 21 (Continued): ZebraLogic Case Study

**Case 2: Continue.**

House 3:
- Possible: Eric (if he is not in House 2 or 1)
- But Eric's PhoneModel = iphone 13, and BookGenre = science fiction.
House 4:
- Possible: Arnold (if he is not in House 3)
- But Arnold's PhoneModel = oneplus 9.
Let's try to place Eric:
Eric must be in a house where:
- PhoneModel = iphone 13
- BookGenre = science fiction
- To the left of Arnold (oneplus 9 user)
Possible positions for Eric:
- If Eric is in House 2: But House 2's BookGenre = fantasy, not science fiction (contradicts Clue 6).
- If Eric is in House 3: Then Arnold must be in House 4 (since 3 < 4).
- If Eric is in House 1: Then Arnold must be in House 2 (but House 2's PhoneModel = samsung galaxy s21, not oneplus 9).
So the only valid position for Eric is **House 3**.
<Omitted.>
**Step5: Let's check if we missed something(confirm):**
Let's re-examine the occupations:
- Occupations: artist, engineer, teacher, doctor
- We have: - House 1: engineer (from Clue 1) - House 3: teacher (Eric) - So remaining: artist and doctor must be in Houses 2 and 4.
House 2: - BookGenre = fantasy (not mystery) - So not the doctor.
House 4: - BookGenre = romance (not mystery) - So not the doctor.
This is a problem. Let's check if we made a mistake.
Let's go back to the initial assumption about the engineer.
Clue 1: The person who is an engineer is directly left of the person who uses a Samsung Galaxy S21.
We assumed the engineer is in House 1 (since only House 1 is directly left of House 2).
<Omitted.>
**Step 6: Verification**
- Clue 1: Engineer (House 1) is directly left of Samsung Galaxy S21 user (House 2) → ✓
- Clue 2: BookGenre = fantasy in House 2 → ✓
- Clue 3: Alice ≠ House 2 → ✓ (Alice is in House 1)
- Clue 4: Eric is a teacher → ✓
- Clue 5: Samsung Galaxy S21 user = fantasy lover → ✓ (House 2)
- Clue 6: iPhone 13 user = science fiction lover → ✓ (Eric, House 3)
- Clue 7: Science fiction lover (House 3) is left of OnePlus 9 user (House 4) → ✓
- Clue 8: Arnold uses OnePlus 9 → ✓
- Clue 9: Doctor = mystery lover → ✓ (Arnold, House 4)
- Clue 10: iPhone 13 user = teacher → ✓ (Eric, House 3)
All clues are satisfied.
**Solution:** {
    "House 1": { "Name": "Alice", "Occupation": "engineer", "BookGenre": "romance", "PhoneModel": "google pixel 6" },
    "House 2": { "Name": "Peter", "Occupation": "artist", "BookGenre": "fantasy", "PhoneModel": "samsung galaxy s21" },
    "House 3": { "Name": "Eric", "Occupation": "teacher", "BookGenre": "science fiction", "PhoneModel": "iphone 13" },
    "House 4": { "Name": "Arnold", "Occupation": "doctor", "BookGenre": "mystery", "PhoneModel": "oneplus 9" }
}

Table 21 (Continued): ZebraLogic Case Study

**Puzzle 2: Qwen3-8B-Chess demonstrated different reasoning steps but failed to solve the problem within the token limit.**

**Puzzle to Solve**:
There are 6 houses, numbered 1 to 6 from left to right, as seen from across the street. Each house is occupied by a different person. Each house has a unique attribute for each of the following characteristics:
- Each person has a unique name: 'Bob', 'Alice', 'Peter', 'Eric', 'Arnold', 'Carol'
- Each mother is accompanied by their child: 'Fred', 'Timothy', 'Samantha', 'Alice', 'Meredith', 'Bella'
- People have unique favorite music genres: 'pop', 'hip hop', 'classical', 'jazz', 'rock', 'country'
- People have unique heights: 'average', 'very tall', 'tall', 'super tall', 'very short', 'short'
**Clues:** 1. There is one house between the person's child is named Samantha and the person who is short.
2. The person's child is named Alice is Bob.
3. The person who loves country music is directly left of Arnold.
4. Alice is the person who is tall.
5. The person who loves pop music is Eric.
6. Bob is somewhere to the right of the person who is super tall.
7. The person's child is named Fred is Peter.
8. The person's child is named Bella is the person who loves hip-hop music.
9. The person who is the mother of Timothy is not in the sixth house.
10. The person who is super tall is somewhere to the right of the person who has an average height.
11. The person's child is named Alice is somewhere to the right of Arnold.
12. There is one house between the person who is short and the person who is very short.
13. The person who is very short is in the fifth house.
14. The person who loves jazz music is not in the fifth house.
15. Carol is somewhere to the left of the person who is the mother of Timothy.
16. The person who is very tall is not in the sixth house.
17. The person who loves classical music is in the sixth house.
18. The person who loves rock music is in the first house.

**Ground-Truth**

```
{
    "House 1": { "Name": "Carol", "Children": "Samantha", "MusicGenre": "rock", "Height": "average" },
    "House 2": { "Name": "Peter", "Children": "Fred", "MusicGenre": "country", "Height": "super tall" },
    "House 3": { "Name": "Arnold", "Children": "Bella", "MusicGenre": "hip hop", "Height": "short" },
    "House 4": { "Name": "Bob", "Children": "Alice", "MusicGenre": "jazz", "Height": "very tall" },
    "House 5": { "Name": "Eric", "Children": "Timothy", "MusicGenre": "pop", "Height": "very short" },
    "House 6": { "Name": "Alice", "Children": "Meredith", "MusicGenre": "classical", "Height": "tall" },
}
```

Table 21 (Continued): ZebraLogic Case Study

**Case 1: Response from Qwen3-8B. The model merely enumerates all clues without engaging in any substantive reasoning. The final answer exhibits multiple inconsistencies with the clues previously stated (highlighted in red).**

**Reasoning**:
Let's break down the clues step by step and deduce the solution logically:
1. From clue 13: The person who is very short is in the fifth house. So, House 5 has Height: very short.
2. From clue 12: There is one house between the person who is short and the person who is very short. Since House 5 is very short, the short person must be in House 3. So, House 3 has Height: short.
3. From clue 10: The person who is super tall is somewhere to the right of the person who has an average height. So, super tall must be in a house after the average height.
4. From clue 16: The person who is very tall is not in the sixth house. So, very tall must be in House 1-5.
5. From clue 17: The person who loves classical music is in the sixth house. So, House 6 has MusicGenre: classical.
6. From clue 18: The person who loves rock music is in the first house. So, House 1 has MusicGenre: rock.
7. From clue 5: The person who loves pop music is Eric. So, Eric's MusicGenre is pop.
8. From clue 3: The person who loves country music is directly left of Arnold. So, Arnold must be in a house to the right of the person who loves country music.
9. From clue 2: The person's child is named Alice is Bob. So, Bob's child is Alice.
10. From clue 7: The person's child is named Fred is Peter. So, Peter's child is Fred.
11. From clue 8: The person's child is named Bella is the person who loves hip-hop music. So, the person who loves hip-hop has a child named Bella.
12. From clue 4: Alice is the person who is tall. So, Alice's Height is tall.
13. From clue 11: The person's child is named Alice is somewhere to the right of Arnold. So, Bob is to the right of Arnold.
14. From clue 15: Carol is somewhere to the left of the person who is the mother of Timothy. So, Carol is to the left of the mother of Timothy.
15. From clue 9: The person who is the mother of Timothy is not in the sixth house. So, the mother of Timothy is in House 1-5.
16. From clue 1: There is one house between the person's child is named Samantha and the person who is short. Since the person who is short is in House 3, the person with child Samantha must be in House 1 or 5. But House 5 is very short, so the person with child Samantha is in House 1.
17. From clue 6: Bob is somewhere to the right of the person who is super tall. So, Bob is in a house after the super tall person.
18. From clue 14: The person who loves jazz music is not in the fifth house. So, jazz is in House 1-4 or 6.
By carefully placing all these constraints, the solution is deduced as follows:
**Solution:** {
    "House 1": { "Name": "Peter", "Children": "Fred", "MusicGenre": "rock", "Height": "super tall" },
    "House 2": { "Name": "Carol", "Children": "Timothy", "MusicGenre": "hip hop", "Height": "average" },
    "House 3": { "Name": "Bob", "Children": "Alice", "MusicGenre": "pop", "Height": "short" },
    "House 4": { "Name": "Eric", "Children": "Samantha", "MusicGenre": "jazz", "Height": "tall" },
    "House 5": { "Name": "Alice", "Children": "Bella", "MusicGenre": "country", "Height": "very short" },
    "House 6": { "Name": "Arnold", "Children": "Meredith", "MusicGenre": "classical", "Height": "very tall" }
}

Table 21 (Continued): ZebraLogic Case Study

**Case 2: Response from Qwen3-8B-Chess. An error was introduced during the intermediate reasoning phase (specifically concerning Bob's house). Crucially, the model exhibited a failure in self-correction, propagating the flawed premise throughout the subsequent steps. The inference was ultimately truncated due to the token limit, preventing the derivation of a final answer.**

Let's tackle this puzzle step by step, using the clues to deduce the correct arrangement.
**Step-by-step Reasoning:**
From Clue 13: - The person who is very short is in the fifth house. - So, House 5 has "Height" = "very short".
From Clue 12: - There is one house between the person who is short and the person who is very short. - Since the very short is in House 5, the short must be in House 3 (since 3 and 5 are separated by one house). <Omitted.>
**Let's start placing known values:**
House 5: - "Height" = "very short" (from Clue 13)
House 1: - "MusicGenre" = "rock" (from Clue 18)
House 6: - "MusicGenre" = "classical" (from Clue 17)
**Now, let's deduce the positions of names and children.**
We know: - "Bob" = "Alice" (child) (from Clue 2) - "Peter" = "Fred" (child) (from Clue 7) - "Bella" = "hip hop" (from Clue 8) - "Eric" = "pop" (from Clue 5) - "Arnold" = "country" is to his left (from Clue 3)
Let's **consider possible positions for Arnold.**
If "country" is directly left of Arnold, then Arnold cannot be in House 1 (no house to the left). So Arnold must be in Houses 2-6.
Let's try placing Arnold in House 3: - Then "country" must be in House 2. Now, "Bob" is to the right of "super tall" (Clue 6), and "Bob" is to the right of Arnold (Clue 11). So if Arnold is in House 3, "Bob" must be in House 4 or 5 or 6.
Also, "super tall" must be to the left of "Bob" (from Clue 6), and "super tall" must be to the left of "average" (from Clue 10). So "super tall" is somewhere in the left part of the grid.
Let's try to place "super tall":
If "super tall" is in House 2, then "Bob" must be in 3, 4, 5, or 6.
But we already have "country" in House 2 (from above), so "super tall" cannot be in House 2.
Try "super tall" in House 3: - Then "Bob" must be in 4, 5, or 6.
But we have "Arnold" in House 3, so "super tall" cannot be in House 3.
**Try "super tall" in House 4: - Then "Bob" must be in 5 or 6.**
This could work.
**Now, "Bob" is in 5 or 6** ×, and "Arnold" is in 3.
Also, "Bob" is to the right of "super tall" (which is in 4), so "Bob" could be in 5 or 6.
Let's assume "super tall" is in House 4, and "Bob" is in House 5.
Now, "Bob" is in House 5, and "Arnold" is in House 3.
Now, "Eric" is in the grid, and "Eric" = "pop" (from Clue 5).
We need to place "Eric" in a house where "pop" is not conflicting with other music genres.
We already have "rock" in House 1, "classical" in House 6, "hip hop" (for Bella) somewhere, "country" in House 2, "pop" for Eric.
So "Eric" must be in a house where "pop" is not already taken.
**Let's try placing "Eric" in House 2:** - But "country" is in House 2 (from above), so this is **not possible.**
Try "Eric" in House 5: - "Bob" is in House 5, so this is not possible.
Try "Eric" in House 6: - "classical" is in House 6, so this is not possible.
So "Eric" must be in House 3 or 4.
But "Arnold" is in House 3, so "Eric" cannot be in House 3.
So "Eric" must be in House 4.
Now, "Eric" is in House 4, and "pop" is his genre.
Try "Eric" in House 4: - But "super tall" is in House 4, so this is not possible.
<Omitted.>
**So "Eric" must be in House 3 or 4.** ×
But "Arnold" is in House 3, so "Eric" cannot be in House 3.
So "Eric" must be in House 4.
**Now, "Eric" is in House 4, and "pop" is his genre.** ×

Table 21 (Continued): ZebraLogic Case Study

**Case 2: Continue**

Now, **let's check if this fits with other clues.**
We have: - "country" in House 2 (left of Arnold in House 3) - "super tall" in House 4 (left of "Bob" in House 5) - "Eric" in House 4 (with "pop")
Now, "Bob" is in House 5, and "Alice" (the person who is tall) is not in House 6 (from Clue 16), so "Alice" must be in House 1-5.
Also, "Alice" = "Bob" (from Clue 2), so "Bob" is "Alice" (the person), and "Alice" is the child of "Bob".
**So "Bob" is a parent, and his child is "Alice".** ✓
Now, "Alice" (the person) must be in a house where she is the parent of "Alice" (the child).
But "Bob" is the parent of "Alice" , so "Alice" (the person) cannot be "Bob".
So "Alice" (the person) must be someone else.
From the list of names: 'Bob', 'Alice', 'Peter', 'Eric', 'Arnold', 'Carol'.
So "Alice" (the person) must be one of these, but not "Bob".
So "Alice" (the person) could be "Carol", "Peter", "Eric", or "Arnold".
But "Eric" is already placed in House 4.
So "Alice" (the person) could be "Carol", "Peter", or "Arnold".
But "Arnold" is already placed in House 3.
So "Alice" (the person) could be "Carol" or "Peter".
Now, **"Alice" (the person) is the one who is tall** ✓ (from Clue 4), so "tall" = "Alice" (the person).
So "Alice" (the person) must be in a house where "tall" is her height.
**Now, let's try to place "Alice" (the person).**
If "Alice" (the person) is in House 2: - Then "tall" = "Alice" (the person), and "country" is in House 2 (from above), so this is possible.
If "Alice" (the person) is in House 3: - "Arnold" is in House 3, so this is not possible.
If "Alice" (the person) is in House 4: - "Eric" is in House 4, so this is not possible.
If "Alice" (the person) is in House 5: - "Bob" is in House 5, so this is not possible.
If "Alice" (the person) is in House 6: - "classical" is in House 6, so this is not possible.
So "Alice" (the person) must be in House 2.
Now, "Alice" (the person) is in House 2, and "tall" = "Alice" (the person).
So "tall" = "Alice" (the person), and "Alice" (the person) is in House 2.
Now, let's check if this fits with other clues.
From Clue 11: - "Bob" (the person) is to the right of Arnold (House 3), so "Bob" is in House 4 or 5 or 6.
We already have "Bob" in House 5.
From Clue 6: - "Bob" is to the right of "super tall" (House 4), so this fits.
From Clue 3: - "country" is in House 2 (left of Arnold in House 3), so this fits.
Now, let's place the remaining names: "Carol", "Peter".
We have "Eric" in House 4, "Arnold" in House 3, "Bob" in House 5, "Alice" (the person) in House 2.
So remaining names: "Carol", "Peter".
These must be in Houses 1 and 6.
From Clue 18: - House 1 has "rock" as music genre.
From Clue 17: - House 6 has "classical" as music genre.
So "rock" is in House 1, "classical" in House 6.
Now, let's place "Carol" and "Peter".
If "Carol" is in House 1: - Then "Peter" must be in House 6.
If "Carol" is in House 6: - Then "Peter" must be in House 1.
Let's try "Carol" in House 1: - Then "Peter" is in House 6.
Now, check other clues.
From Clue 7: - "Peter" = "Fred" (child), so "Fred" is the child of "Peter" (in House 6).
From Clue 8: - "Bella" = "hip hop", so "Bella" is the child of someone who loves "hip hop".
<Omitted.>
Now, let's assume
**End from here because of token limit.**

Table 21 (Continued): ZebraLogic Case Study

**Puzzle 3: Qwen3-8B-Chess similarly displayed an extensive number of reasoning steps. Crucially, it detected an initial error and subsequently engaged in a self-correction mechanism (or procedural adjustment).**

**Puzzle to Solve**
There are 5 houses, numbered 1 to 5 from left to right, as seen from across the street. Each house is occupied by a different person. Each house has a unique attribute for each of the following characteristics:
- Each person has a unique name: 'Peter', 'Alice', 'Bob', 'Eric', 'Arnold'
- The people are of nationalities: 'norwegian', 'german', 'dane', 'brit', 'swede'
- People have unique favorite book genres: 'fantasy', 'biography', 'romance', 'mystery', 'science fiction'
- Everyone has something unique for lunch: 'stir fry', 'grilled cheese', 'pizza', 'spaghetti', 'stew'
- Each person has a favorite color: 'red', 'green', 'blue', 'yellow', 'white'
- The people keep unique animals: 'bird', 'dog', 'cat', 'horse', 'fish'

**Clues:**
1. The person who loves fantasy books is the Norwegian.
2. The cat lover and the person who loves biography books are next to each other.
3. The German is Bob.
4. The person who loves yellow is Bob.
5. The person whose favorite color is green is Peter.
6. There is one house between the Dane and the person who is a pizza lover.
7. The person who loves blue is somewhere to the left of the Dane.
8. The person who loves eating grilled cheese is somewhere to the left of the Norwegian.
9. The person who loves the spaghetti eater is Peter.
10. The person who keeps horses is Alice.
11. The fish enthusiast is directly left of the person who loves science fiction books.
12. There is one house between the Norwegian and Arnold.
13. The person who loves romance books is the British person.
14. There are two houses between the Norwegian and Alice.
15. The bird keeper is the person whose favorite color is red.
16. The dog owner is directly left of the fish enthusiast.
17. The person who loves the stew is the Norwegian.

**Ground-truth**

```
{
    "House 1": {"Name": "Bob", "Nationality": "german", "BookGenre": "mystery", "Food":
"grilled cheese", "Color": "yellow", "Animal": "dog"},
    "House 2": {"Name": "Eric", "Nationality": "norwegian", "BookGenre": "fantasy",
"Food": "stew", "Color": "blue", "Animal": "fish"},
    "House 3": {"Name": "Peter", "Nationality": "dane", "BookGenre": "science fiction",
"Food": "spaghetti", "Color": "green", "Animal": "cat"},
    "House 4": {"Name": "Arnold", "Nationality": "swede", "BookGenre": "biography",
"Food": "stir fry", "Color": "red", "Animal": "bird"},
    "House 5": {"Name": "Alice", "Nationality": "brit", "BookGenre": "romance", "Food":
"pizza", "Color": "white", "Animal": "horse"}
}
```

Table 21 (Continued): ZebraLogic Case Study

**Case 1: Response from Qwen3-8B.**

**Reasoning:** Let's start by analyzing the clues step by step to deduce the attributes for each house.
1. From Clue 17, the Norwegian loves stew.
2. From Clue 8, the grilled cheese lover is to the left of the Norwegian.
3. From Clue 12, there is one house between the Norwegian and Arnold.
4. From Clue 14, there are two houses between the Norwegian and Alice.
5. From Clue 10, Alice keeps horses.
6. From Clue 3, the German is Bob.
7. From Clue 4, Bob loves yellow.
8. From Clue 5, Peter loves green.
9. From Clue 9, Peter loves spaghetti.
10. From Clue 13, the British person loves romance books.
11. From Clue 1, the Norwegian loves fantasy books.
12. From Clue 15, the bird keeper loves red.
13. From Clue 16, the dog owner is directly left of the fish enthusiast.
14. From Clue 11, the fish enthusiast is directly left of the science fiction lover. 15. From Clue 2, the cat lover and the biography lover are next to each other.
16. From Clue 6, there is one house between the Dane and the pizza lover.
17. From Clue 7, the blue lover is to the left of the Dane.
Using these clues, we can deduce the following:
- The Norwegian must be in House 3 (since they are between the grilled cheese lover and Arnold, and two houses away from Alice).
- The grilled cheese lover is in House 2.
- Arnold is in House 4.
- Alice is in House 1 (two houses away from the Norwegian).
- Bob is the German and loves yellow.
- Peter is in House 5, loves green, and spaghetti.
- The British person (who loves romance books) is in House 2.
- The Dane is in House 4.
- The fish enthusiast is in House 5, and the science fiction lover is in House 6 (but since there are only 5 houses, this is invalid, so we must adjust).
- Adjusting, the fish enthusiast is in House 4, and the science fiction lover is in House 5.
- The dog owner is in House 3, and the fish enthusiast is in House 4.
- The cat lover and biography lover are next to each other.
- The bird keeper loves red and is in House 1.
- The Dane is in House 4, and the pizza lover is in House 3.
- The blue lover is in House 3, and the Dane is in House 4.
Putting it all together, we get the final arrangement.
**Solution:** {
    "House 1": { "Name": "Alice", "Nationality": "Norwegian", "BookGenre": "Fantasy", "Food": "Grilled Cheese", "Color": "Red", "Animal": "Bird" },
    "House 2": { "Name": "Peter", "Nationality": "Brit", "BookGenre": "Romance", "Food": "Pizza", "Color": "Green", "Animal": "Cat" },
    "House 3": { "Name": "Bob", "Nationality": "German", "BookGenre": "Biography", "Food": "Stir Fry", "Color": "Blue", "Animal": "Dog" },
    "House 4": { "Name": "Arnold", "Nationality": "Dane", "BookGenre": "Science Fiction", "Food": "Stew", "Color": "White", "Animal": "Fish" },
    "House 5": { "Name": "Eric", "Nationality": "Swede", "BookGenre": "Mystery", "Food": "Spaghetti", "Color": "Yellow", "Animal": "Horse" }
}

Table 21 (Continued): ZebraLogic Case Study

**Case 2: Response from Qwen3-8B-Chess.**

Let's tackle this step by step. First, let's list out all the clues and see what we can deduce:

**Clues:** 1. Fantasy books lover is Norwegian. 2. Cat lover and biography books lover are adjacent. 3. German is Bob. 4. Yellow color lover is Bob. 5. Green color lover is Peter. 6. One house between Dane and pizza lover. 7. Blue color lover is left of Dane. 8. Grilled cheese lover is left of Norwegian. 9. Spaghetti eater is Peter. 10. Horses lover is Alice. 11. Fish enthusiast is left of science fiction books lover. 12. One house between Norwegian and Arnold. 13. Romance books lover is British. 14. Two houses between Norwegian and Alice. 15. Bird keeper has red color. 16. Dog owner is left of fish enthusiast. 17. Stew lover is Norwegian.

**Initial Deductions:** - From clue 17: Norwegian's food is stew. - From clue 9: Peter's food is spaghetti. - From clue 5: Peter's color is green. - From clue 13: British person's book genre is romance. - From clue 10: Alice's animal is horse. - From clue 15: Bird keeper's color is red. - From clue 4: Bob's color is yellow. - From clue 3: Bob's nationality is German. - From clue 1: Norwegian's book genre is fantasy.

**Let's start placing what we know:**

Nationalities: - Norwegian: from clue 17, food is stew. - German: Bob (clue 3) - British: from clue 13, book genre is romance. - Dane: not yet placed. - Swede: not yet placed.

Names: - Peter: color green (clue 5), food spaghetti (clue 9) - Bob: nationality German (clue 3), color yellow (clue 4) - Alice: animal horse (clue 10) - Arnold: not yet placed. - Eric: not yet placed.

Book Genres: - Fantasy: Norwegian (clue 1) - Romance: British (clue 13) - Biography: not yet placed. - Mystery: not yet placed. - Science Fiction: not yet placed.

<Omitted.>

Clue 14: Two houses between Norwegian and Alice. - Norwegian and Alice are not adjacent, not one apart. **Possible positions: (1,3), (2,4), (3,5). ×**

Norwegian in house 4: - Grilled cheese lover must be in 3. - From clue 12: Arnold is in 2 or 6 (6 invalid). So Arnold in 2. - From clue 14: Alice is in 2 or 6 (6 invalid). So Alice in 2. But 2 is Arnold. Invalid.

Norwegian in house 5: - Grilled cheese lover must be in 4. - From clue 12: Arnold is in 3 or 7 (7 invalid). So Arnold in 3. - From clue 14: Alice is in 3 or 7 (7 invalid). So Alice in 3. But 3 is Arnold. Invalid.

So the only valid position for Norwegian is house 3.

<Omitted.>

So Dane must be in 5.

Then, blue color lover must be in 4 (Peter, color green). Doesn't work.

**So this is a problem.**

Maybe I need to **adjust**.

Let's go back to Norwegian in 3, Arnold in 5, Alice in 1.

This is Option 2.

Option 2: Norwegian in 3, Arnold in 5, Alice in 1.

- House 3: Norwegian, food stew, book genre fantasy. - House 5: Arnold. - House 1: Alice, animal horse.

Now, clue 8: Grilled cheese lover is left of Norwegian (3). So must be in 2.

House 2: Grilled cheese lover.

Now, let's place Peter.

Peter's food is spaghetti (clue 9), color is green (clue 5).

Possible houses: 4.

House 4: Peter, food spaghetti, color green.

Now, Bob's nationality is German (clue 3), color is yellow (clue 4).

Possible houses: 2 (grilled cheese), 4 (Peter), 5 (Arnold).

So Bob must be in 2.

House 2: Bob, nationality German, color yellow, food grilled cheese.

Now, clue 7: Blue color lover is left of Dane.

Possible positions:

If Dane is in 4, then blue color lover in 3 (Norwegian, color not yet known).

If Dane is in 5, then blue color lover in 4 (Peter, color green). Doesn't work.

If Dane is in 2, then blue color lover in 1 (Alice, color not yet known).

Table 21 (Continued): ZebraLogic Case Study

**Case 2: Continue**

Let's check if this works.
Let's assume Dane is in 4.
Then, blue color lover is in 3 (Norwegian).
So Norwegian's color is blue.
That works.
Now, let's check clue 15: Bird keeper has red color.
Possible houses: 1 (Alice, color not yet known), 2 (Bob, color yellow), 3 (Norwegian, color blue), 4 (Peter, color green), 5 (Arnold, color not yet known).
So bird keeper must be in a house with red color.
Possible: 5 (Arnold).
So Arnold's color is red.
That works.
Now, let's place the remaining:
- Dane is in 4. - Norwegian in 3 (color blue). - Bob in 2 (color yellow, food grilled cheese). - Peter in 4 (color green, food spaghetti). Wait, 4 is Dane. So Peter cannot be in 4.
So Peter must be in 5.
House 5: Peter, color green, food spaghetti.
But 5 is Arnold. So this is invalid.
**So this is a problem.**
Hmm, this is getting complex. Let's try to **use a table** to organize.
Let's create a table:
| House | Name | Nationality | Book Genre | Food | Color | Animal |
|——-|——|————|————|———|——|——|
| 1 | Alice | ? | ? | ? | ? | Horse |
| 2 | Bob | German | ? | Grilled Cheese | Yellow | ? |
| 3 | ? | Norwegian | Fantasy | Stew | ? | ? |
| 4 | ? | ? | ? | ? | ? | ? |
| 5 | Arnold | ? | ? | ? | ? | ? |
Now, let's fill in what we know.
Clue 10: Alice's animal is horse (house 1).
Clue 3: Bob is German (house 2).
Clue 4: Bob's color is yellow (house 2).
Clue 17: Norwegian's food is stew (house 3).
Clue 1: Norwegian's book genre is fantasy (house 3).
Clue 8: Grilled cheese lover is left of Norwegian (house 2).
Now, let's place Peter.
Peter's food is spaghetti (clue 9), color is green (clue 5).
<Omitted.>
But clue 11: Fish enthusiast is left of science fiction books lover.
So science fiction books lover must be in 4.
That works.
Now, let's check clue 11: Fish enthusiast (3) is left of science fiction books lover (4).
Yes.
Now, what about clue 2: Cat lover and biography books lover are adjacent.
**End from here because of token limit.**

## G.4 LEGAL MOVES AS POTENTIAL CONSTRAINTS

In the aforementioned evaluation, we observed superior model performance under the "with legal moves" configuration, which aligns with theoretical expectations. This approach enables the model to select exclusively from predetermined legal moves, thereby virtually eliminating the possibility of generating illegal moves. Barring fundamental deficiencies in instruction-following capabilities, models can consistently select appropriate moves from the provided legal move set. However, this

configuration may inadvertently constrain the model's full potential. We discovered that under the "with legal moves" condition, models frequently exhibit diminished analytical engagement, characterized by the tendency to bypass deliberative reasoning processes and directly output moves without substantive thought. Conversely, when the same board state (FEN) is evaluated under "without legal moves" constraints, models often generate superior move selections. Under these unrestricted conditions, models engage in comprehensive tactical analysis, systematically considering viable moves for each piece and conducting thorough evaluations of all possible move qualities. This phenomenon manifests consistently across various non-reasoning models, including Qwen3-235b-a22b (non-thinking) and Deepseek-V3. Illustrative examples of this behavioral pattern are presented in Table 22 and Figure 8.

Table 22: With legal moves, the model's potential may not be fully realized

**Case 1: Qwen3 outputs a move directly (legal moves provided)**

**Query:**
**System prompt is omitted.**
User:
Current board position in FEN notation: rnbqkbnr/pp2pppp/3p4/8/3pP3/5N2/PPP2PPP/ RNBQKB1R w KQkq - 0 4
Legal moves in UCI notation: f3g5, f3e5, f3h4, f3d4, f3d2, f3g1, h1g1, f1a6, f1b5, f1c4, f1d3, f1e2, e1e2, e1d2, d1d4, d1d3, d1e2, d1d2, c1h6, c1g5, c1f4, c1e3, c1d2, b1c3, b1a3, b1d2, e4e5, h2h3, g2g3, c2c3, b2b3, a2a3, h2h4, g2g4, c2c4, b2b4, a2a4

**Response from Qwen3-235b-a22b(Without any reasoning process, providing its answer directly, exhibiting certain lazy behavior)**
```
e4e5
```

**Case 2: Qwen3 thinks carefully and outputs a better move (no legal moves provided)**

Table 22 (Continued): With legal moves, the model's potential may not be fully realized

**Query:**
**System prompt is omitted.**
User:
Current board position in FEN notation: rnbqkbnr/pp2pppp/3p4/8/3pP3/5N2/PPP2PPP/
RNBQKB1R w KQkq - 0 4

**Response from Qwen3-235b-a22b**
Let's analyze the position carefully.

**Position in FEN:** rnbqkbnr/pp2pppp/3p4/8/3pP3/5N2/PPP2PPP/RNBQKB1R w KQkq - 0 4
**Interpretation:**
- Side to move: White
- Opening: This looks like a **Scotch Game** or a variation of it, with an early d4 by Black.
- Material is equal
- White has a central pawn on e4, and Black has pawns on d6 and d4, indicating a possible gambit or aggressive central play by Black.
- White has not yet developed much, but has the advantage of a tempo lead.
- Black's queen is still on d8, and Black's kingside is intact.
- White's knight is already on f3, and the e4 pawn is protected.
- White's light-squared bishop and queen are still on their starting squares.

**Candidate Moves:**
1. **e4-e5** – Pushing the pawn to open the center and gain space.
2. **Qxd4** – Capturing the pawn with the queen.
3. **Nxd4** – Capturing the pawn with the knight.
4. **Bc4** – Developing with a strong bishop eyeing f7.
5. **c3** – Supporting a later d4 or preparing for development.

**Evaluation:**
- **e5** is a strong move, gaining space and limiting Black's options in the center.
- **Qxd4** is playable, but exposes the queen early and may lead to early exchanges.
- **Nxd4** is also playable, but after **Qxd4**, White must move the knight or queen again, giving Black a tempo advantage.
- **Bc4** is a good developing move, preparing castling and exerting pressure on the kingside.
- **c3** is a bit slow here; White should focus on development and center control.

**Best Plan:**
Developing with **Bc4** is the most natural and strong move. It develops a piece, pressures the f7 square (which is always a weak point in the opening), and keeps all options open for White in the center.

**Final Move:**
```
f1c4
```

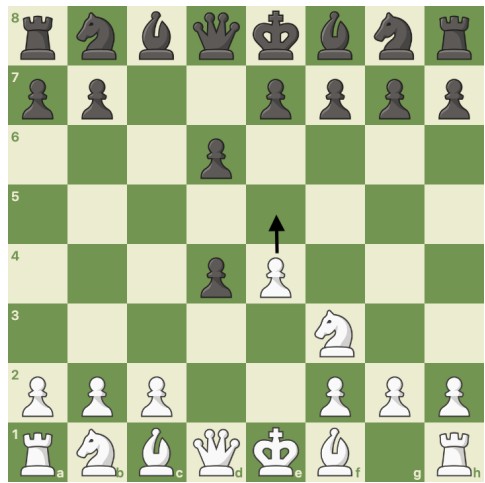
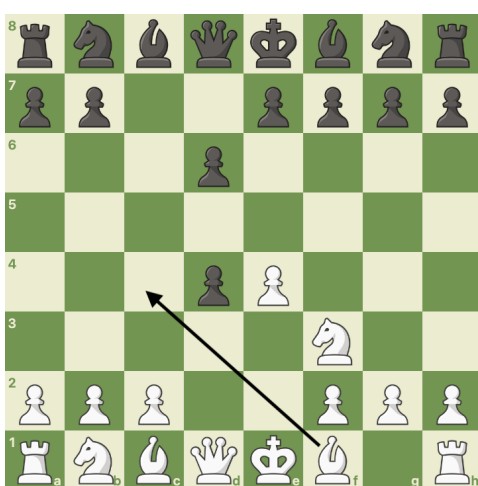

(a) Move from Qwen3 when providing legal moves. Qwen3 choose e4e5. It's a normal move.

(b) Move from Qwen3 when not providing legal moves. After careful consideration, Qwen3 choose f1f4. This move keeps options open for White in the center.

Figure 8: Providing legal moves may lock model's potential

## G.5 RL TRAINING ANALYSIS

**Training Curve**  Our RL training mean-reward and response length are shown in Figure 9, which were downloaded from WanDB[6]. Both mean reward and response length exhibit upward trends, indicating that the model's chess strategic reasoning capabilities are progressively improving.

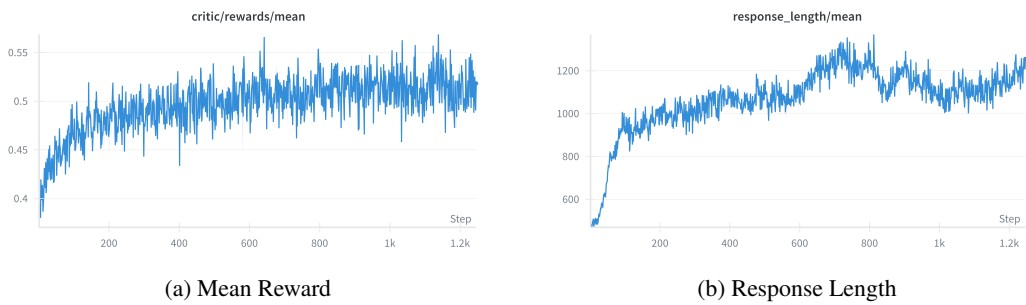

(a) Mean Reward                                    (b) Response Length

Figure 9: Mean Reward and Response Length Curve of RL Training

**Rollout**  We conducted a comprehensive analysis of the evolution in Qwen3-8B-Chess's board evaluation reasoning throughout reinforcement learning training. By examining the model's cognitive processes for identical chess positions across different training phases, we traced the development of its analytical capabilities. Cases 1 and 2 in Table 23 exemplify this progression. During the early training phase, while Qwen3-8B-Chess demonstrated fundamentally sound analysis, its strategic depth remained limited. Specifically, the model should have prioritized material capture over king threats, as such threats lacked tactical substance in the given position. In contrast, during the late training phase, the model exhibited a marked preference for material acquisition and engaged in systematic verification processes, critically evaluating the validity of its own reasoning. Furthermore, in Case 3 of Table 23, Qwen3-8B-Chess demonstrated enhanced analytical rigor through dual verification mechanisms and extended tactical calculations. The model began simulating move sequences systematically, employing this methodological approach to identify what it determined to be the optimal solution.

---
[6] https://wandb.ai/

Table 23: Qwen3-8B-Chess thought difference between RL training early stage and late stage

**Case 1: Qwen3-8B thought process of early training stage**

**Query:**
System prompt is omitted.
User:
Current board position in FEN notation: 4k3/2R5/8/P6K/2p4b/2P2P2/P7/8 w - - 5 69
Legal moves in UCI notation: c7c8, c7h7, c7g7, c7f7, c7e7, c7d7, c7b7, c7a7, c7c6, c7c5, c7c4, h5h6, h5g6, h5h4, h5g4, a5a6, f3f4, a2a3, a2a4

**Response from Qwen3-8B-Chess in early RL training stage**

Let's analyze the position carefully.
**FEN:** 4k3/2R5/8/P6K/2p4b/2P2P2/P7/8 w - - 5 69
This is a late-middlegame or early-endgame position. White has a rook on c7, a king on h5, pawns on a5, c3, f3, and a2. Black has a king on e8, a bishop on h4, and a pawn on c4.
**Key Observations:**
- White Rook on c7 is active and can potentially promote if supported.
- Black Bishop on h4 is currently attacking the pawn on g3 (which doesn't exist yet), but more importantly, it's eyeing the long diagonal.
- Black King on e8 is somewhat passive, but could become active depending on how the game unfolds.
- White King on h5 is quite advanced and could support a pawn push on the kingside.
- Pawn Structure:
- White has a passed pawn on a5, which could be pushed.
- Black has a pawn on c4, which is isolated and potentially weak.
- White has pawns on c3 and f3, which are mostly defensive.
- White has a potential passed pawn on a2, but it's not very advanced.
**Candidate Moves:**
From the list of legal moves:
- Rook moves from c7:
- 'c7c8': Attacks the black king directly. Could lead to a check.
- 'c7h7', 'c7g7', 'c7f7', 'c7e7', 'c7d7', 'c7b7', 'c7a7': Various rook moves along the 7th rank.
- 'c7c6', 'c7c5', 'c7c4': Moving the rook down the file.
- King moves from h5:
- 'h5h6', 'h5g6', 'h5h4', 'h5g4': King moves toward the center or supports pawn advances.
- Pawn moves: - 'a5a6': Pushing the a-pawn forward.
- 'f3f4': Pushing the f-pawn.
- 'a2a3', 'a2a4': Pushing the a-pawn from a2.
**Evaluation:**
The most forcing move is 'c7c8', which puts the rook on the same file as the black king and gives check. This forces the black king to move, and White can then possibly follow up with more active play.
Let's look at that line:
1. 'c7c8+' (Rook checks the king) - Black must move the king: 'Ke8-d7' or 'Ke8-e7'
2. Then White can continue to press with the rook, possibly supporting the a-pawn or advancing the king.
Alternatively, moving the king ('h5g6') could also be useful to support the center, but it's less immediate.
Pushing the a-pawn ('a5a6') is also a good idea, but it's slower and doesn't create an immediate threat.
**Best Move:**
The most active and forcing move is:
```
c7c8
```

Table 23 (Continued): Qwen3-8B-Chess's thought during RL training

**Case 2: Qwen3-8B thinks with confirm process.**

---

**Response from Qwen3-8B-Chess in the late RL training stage**

Let's analyze the position carefully.

**FEN:** 4k3/2R5/8/P6K/2p4b/2P2P2/P7/8 w - - 5 69

This is a late-middlegame or early-endgame position. White has a rook on c7, a king on h5, pawns on a5, c3, f3, a2. Black has a king on e8, a bishop on h4, and a pawn on c4.

**Key Observations:**

- White's rook is active on c7, but not yet fully optimized.
- Black's bishop is active on h4, controlling some diagonals.
- Black's king is in the center, but not immediately vulnerable.
- White's king is advanced on h5, which could be useful for supporting pawn advances or attacking.
- Pawn structure: White has a passed pawn on a5, and Black has a passed pawn on c4. Both sides have pawns that could become targets.

**Candidate Moves:**

From the list of legal moves: - 'c7c8': Rook moves to c8 (could be useful if aiming to control the back rank)
- 'c7h7': Rook moves to h7 (could support king on h5)
- 'c7g7': Rook moves to g7 (could control g-file)
- 'c7f7': Rook moves to f7 (could control f-file)
- 'c7e7': Rook moves to e7 (could control e-file)
- 'c7d7': Rook moves to d7 (could control d-file)
- 'c7b7': Rook moves to b7 (could control b-file)
- 'c7a7': Rook moves to a7 (could support a-pawn)
- 'c7c6': Rook moves to c6 (could attack c4 pawn)
- 'c7c5': Rook moves to c5 (could control c-file)
- 'c7c4': Rook captures on c4 (could win a pawn)
- 'h5h6': King moves forward (could support pawn advance)
- 'h5g6': King moves toward center (could be risky)
- 'h5h4': King captures bishop (if bishop is not protected)
- 'h5g4': King moves toward center (could be risky)
- 'a5a6': Pawn advances (could create a passed pawn)
- 'f3f4': Pawn advances (could control center)
- 'a2a3': Pawn advances (could support a5 pawn)
- 'a2a4': Pawn advances (could create a passed pawn)

**Evaluation:**

Most Promising Moves: - 'c7c4': Capturing the pawn on c4 would give White a material advantage (rook for pawn). But we need to check if it's safe.
- 'h5h4': Capturing the bishop on h4. This would be a major material gain (king captures bishop).

**But is the bishop protected?** Let's see: - After 'h5h4', Black can respond with 'g5g4' (if there's a pawn on g5), but there isn't. So the bishop is not protected. So this is a free bishop. So 'h5h4' is a **very strong move**, winning a bishop.

**Let's confirm:** - Is the bishop on h4 protected? No.
- Is the king on h5 safe after capturing? Yes, because the bishop is not protected, and no immediate threats.

So 'h5h4' is a **free bishop**, which is a major material advantage.

**Conclusion:**

The best move is to capture the bishop with the king.
```
h5h4
```

---

**Case 3: Qwen3-8B thinks with some double-check and moving sequence analysis process.**

Table 23 (Continued): Qwen3-8B-Chess's thought during RL training

Let's analyze the position carefully.
**FEN:** 2bq2k1/5p2/p4rn1/1pb1pNQ1/2p1P1P1/2P5/PPB2PN1/4R1K1 w - - 1 30
You are playing as **White**.
**Key Observations:**
**King Safety:**
- White king is on g1, relatively safe for now. Black king is on g8, also fairly safe.
**Piece Activity:**
- White's knight on f5 is active. White's queen on g5 is active and could be dangerous. Black's knight on g6 is defending h8 and f8 squares. Black's bishop on c5 is active. Black's rook on f6 is active. White's bishop on c2 is somewhat passive but can become active. White's rook on e1 is on an open file (e-file).
**Pawn Structure:**
- White has a pawn on e4, which is advanced. Black has a pawn on c4, which is also advanced. There are some isolated and potential weak pawns.
**Threats:**
- White's queen on g5 is eyeing the g-file and could potentially threaten the black king if supported. The knight on f5 could jump to d6 or h6. Black's knight on g6 is protecting the king and may be involved in defense. Black's bishop on c5 is targeting the e3 square and could be part of a diagonal.
**Candidate Moves:**
Let's look at some promising moves for White:
**1. f5d6**
- Knight moves to d6, attacking the bishop on c8 and possibly threatening to fork. Also attacks the pawn on e5. Could disrupt Black's structure.
**2. f5h6**
- Knight moves to h6, attacking the rook on f6. Could lead to a tactical opportunity if the rook moves. But Black can just move the rook.
**3. g5g6**
- Queen moves to g6, attacking the knight on g6. If knight moves, White can possibly capture on f6. But Black can play ...Rh8 or ...Qe8 to defend.
**4. c2a4**
- Bishop moves to a4, targeting the pawn on c4. Could put pressure on Black's queenside.
**5. f5g7**
- Knight moves to g7, attacking the pawn on g7. But that's not useful since it's a pawn and not a piece.
**6. g5f6**
- Queen captures rook on f6! This is a very tempting move.
**Let's check if it's safe:**
- After 'g5f6', Black must recapture with the knight ('g6f6').
- Then White can play 'f5d6', attacking the bishop on c8 and threatening to win material.
- Also, White gains a rook for a queen? No, wait: Queen takes rook, then knight takes queen.
- So White wins a rook and loses a queen? That's a bad trade.
- Unless there's a follow-up.
**Wait, let's double-check:**
- 'g5f6' (Queen takes rook). Black must respond with 'g6f6' (knight takes queen). Now White can play 'f5d6', attacking the bishop on c8 and the pawn on e5. Bishop on c8 is defended by queen (which is gone), so not immediately vulnerable. But White now has a knight on d6, which is strong.
But is this a good trade? Queen for rook and knight?
**Yes! Because Black's knight was defending the king, and now it's gone.** Also, White's knight on d6 is very active.
So this seems like a **good tactical opportunity**.
Let's confirm: 'g5f6' (Qxf6) ->'g6f6' (NxQ) -> 'f5d6' (Ng6-e5 or Ng6-c7?)
But after 'f5d6', Black can play 'c8d7' (Bxd7), and White can play 'd6e8' (Nxe8), winning the queen!
**Wait, let's walk through:**
1. 'g5f6' (Qxf6) 2. 'g6f6' (NxQ) 3. 'f5d6' (Ng6-e5 or Ng6-c7?) 4. 'c8d7' (Bxd7) 5. 'd6e8' (Nxe8) —
White wins the queen!
**Yes! That's a winning sequence.**
So the move 'g5f6' leads to a forced win of the queen.
Is there any way Black can avoid this? No, because after 'g5f6', Black must take with the knight. If they don't, White keeps the rook.
So the best move is:
``` g5f6 ```
This is a tactical shot that wins material and gives White a clear advantage.

