# OpenReview forum: "ChessArena: A Chess Testbed for Evaluating Strategic Reasoning Capabilities of Large Language Models"
_ICLR.cc/2026/Conference — ICLR 2026 Conference Withdrawn Submission_

### Official Review · Reviewer_UYks · 2025-10-26

**Soundness:** 3
**Presentation:** 3
**Contribution:** 3
**Rating:** 4
**Confidence:** 4

**Summary:**

This paper introduces ChessArena, a competitive platform where large language models play chess against each other to evaluate their strategic reasoning capabilities. The framework supports four play modes (Bullet, Blitz, Standard, Blindfold), uses a Glicko rating system with optimized competition sampling, and includes three fine-grained evaluation tasks (basic understanding, move selection, puzzle solving). The authors evaluate 13+ LLMs through 800+ games and find that no model defeats Maia-1100, a chess engine at human amateur level. They also present Qwen3-8B-Chess, trained via supervised fine-tuning and GRPO reinforcement learning, showing improved performance on chess tasks and some transfer to logical reasoning benchmarks.

**Strengths:**

The paper demonstrates careful experimental design with systematic evaluation across multiple state-of-the-art models and substantial computational investment in running hundreds of games. The competition sampling algorithm is mathematically grounded with clear derivations in the appendix. The fine-grained evaluation tasks successfully isolate different aspects of chess ability, and the diagnostic value of distinguishing format errors from illegal moves from suboptimal legal moves is useful for understanding model failures. The commitment to open-source the platform, competition data, and trained model would benefit the community. The finding that thinking models generally outperform non-thinking models is interesting, and the observation that providing legal moves can paradoxically reduce analytical depth (Appendix G.3) merits further investigation.

**Weaknesses:**

The fundamental issue is the mismatch between method choice and task nature. Chess is a sequential decision problem where strategic value emerges from multi-move combinations—classic tactics like "sacrifice for positional advantage" require evaluating positions 3-5 moves ahead. Yet the paper uses single-step RL, sampling board states independently and scoring individual moves via Stockfish. This design cannot learn sequence-level strategy: a move appearing unfavorable now (e.g., sacrificing material) may lead to checkmate later, but receives negative reward under single-step evaluation. The paper provides no explanation for not using PPO (supporting full trajectory learning) or MCTS (the gold standard in chess AI), nor discusses why self-play was not explored despite the competition framework already supporting model-vs-model games.
The GRPO choice lacks justification. The paper cites its success on mathematical reasoning, but math verification is single-step (answer correctness), whereas chess evaluation is inherently sequential (outcome of move sequences). Reducing chess to "selecting from candidates" obscures genuine strategic reasoning—generating and evaluating complex move sequences. Experimental data confirms this: models perform significantly better with legal move lists provided (MAR +41.1% vs -1.2%), indicating they learn ranking rather than reasoning.
The reward design further exposes methodological flaws. Using binary top-3 indicators discretizes Stockfish's continuous evaluation (0-100% win rate) into {0,1}, losing nuanced quality distinctions. Moves with +50% and +48% win rates receive identical rewards, eliminating incentive for fine-grained judgment. The 56,000 training samples are called "virtually unlimited," yet chess has ~10^43 legal positions—this scale is negligible in the search space.
The Blindfold mode claims to test multi-turn reasoning, but this is a false premise in perfect information games. The current FEN string fully encodes all decision-relevant information; move history is theoretically redundant. Providing "the last 10 moves to prevent fivefold repetition" actually reveals models cannot accurately parse FEN representations—this tests format parsing, not strategic memory. Case studies (Table 15) confirm: thinking models fail at board reconstruction, non-thinking models use shortcuts based on recent moves, neither demonstrates genuine state reasoning.
Generalization experiments lack basic controls. Table 16 shows ZebraLogic improves 17.6%, but six of eight benchmarks decline, with code tasks systematically degrading. The paper assumes chess training transfers strategic reasoning, but fails to exclude alternative explanations: the format reward in RL training (ε_f=0.1) itself incentivizes structured outputs; the "Step 1-6" framework and "Verification" steps appearing after training (Table 17) may purely result from format constraints rather than chess knowledge. Essential ablation experiments—RL training on equivalent-scale math or logic problems as controls—are completely absent.

**Questions:**

The adoption of single-step RL over sequential decision methods requires explicit justification. Standard chess AI (AlphaZero, Leela Chess) relies on MCTS to evaluate future move trees, while algorithms like PPO naturally support learning from complete game trajectories. Your 56,000 board states are extracted from Lichess games, which inherently provide full (state, action, next_state, ..., outcome) sequences. Why reduce these to independent single-step samples? If the technical reason is that GRPO implementation is simpler, can you provide experimental evidence that single-step methods are sufficiently effective for chess? If the conceptual reason is that single-step evaluation captures chess strategy, this contradicts 50 years of domain research (from minimax to MCTS) and requires strong argumentation.
The absence of self-play is puzzling. You have implemented a complete competition framework supporting any two models playing against each other—precisely the infrastructure needed for self-play. The standard pipeline is: SFT initialization → self-play RL → iterative improvement. The paper stops at SFT, training only on static Lichess positions. Can you clarify whether self-play was considered? If the issue was weak base models (Qwen3-8B's >70% illegal move rate), did you attempt bootstrapping with high-quality model (e.g., GPT-4.1) game data before transitioning to self-play? The current approach relies on Stockfish-labeled static data, limiting exploration space—the model can never discover novel strategies Stockfish hasn't evaluated.
Can the reward function be continuous? Stockfish outputs precise win-rate evaluations (e.g., +0.5 for slight advantage, +3.0 for decisive advantage), but the paper binarizes this to top-3 indicators. This means the 3rd and 4th ranked moves (potentially differing by 0.1% win rate) receive completely different rewards (1 vs 0), while the 1st and 3rd ranked moves (potentially differing by 5%) receive identical rewards. Sparse rewards are known to reduce RL training efficiency—why not use reward = normalize(stockfish_score)? If concerned about continuous reward scaling, were rank-based continuous rewards attempted, such as reward = 1 - rank/num_legal_moves?
Generalization experiments need critical controls. The current design compares chess RL vs no RL, but cannot distinguish chess-specific effects from general RL effects. Suggested ablations: (1) GRPO training on 56k math reasoning problems, test on ZebraLogic; (2) GRPO training on 56k code problems, same test; (3) training on mixed-domain samples. If (1)(2) also show similar ZebraLogic improvements, chess's specificity is questionable. Additionally, can the case study in Table 17 be expanded? Currently only one success case exists—we need to know: (a) qualitative differences between success/failure cases before/after training on ZebraLogic; (b) whether improvements systematically manifest as better constraint tracking and logical chain completeness, or are random; (c) whether counterexamples exist where outputs become more structured but answers remain incorrect.
How do you explain the performance degradation from SFT-Stage2 (6/8 benchmarks decline)? This model becomes the RL starting point, meaning final improvements may partially be "repairing SFT damage" rather than "exceeding baseline." Can you provide a direct Base → Base+RL comparison, skipping SFT-Stage2? Or explain why chess domain data (distilled from GPT-4.1, etc.) so severely harms code generation ability (CruxEval 73.25→68.0)? This is important for understanding the cost of domain adaptation.

---

> ### Author Response · Authors · 2025-11-26
> **Response to Reviewer UYks**
>
> We sincerely thank Reviewer for the insightful questions and valuable suggestions. We appreciate your technical enthusiasm and provide the following responses after carefully reviewing your comments.
>
> First, we would like to clarify that **ChessArena** is designed as an evaluation **benchmark** primarily focused on assessing the strategic reasoning capabilities of Large Language Models (LLMs) through head-to-head competition. We also designed fine-grained evaluation metrics to decouple the various aspects of model performance in chess strategy and reasoning.
>
> Observing the performance issues of models on ChessArena, we initiated a process of model distillation and curated a domain-specific Chess Supervised Fine-Tuning (SFT) dataset. Subsequently, we performed a chess post-training step on a smaller model, Qwen3-8B, to establish it as a relatively strong **baseline** within the ChessArena environment—which we refer to as Qwen3-8B-Chess.
>
> We acknowledge that Qwen3-8B-Chess is still far from achieving SOTA performance among specialized Chess LLMs. We recognize that many other training methodologies, such as the **self-play** approach you mentioned, could potentially further enhance its capability. Due to resource constraints, which prevented long-context post-training, we primarily focused on single-step SFT and Reinforcement Learning (RL) to train Qwen3-8B-Chess.
>
> We are very grateful for your highly valuable suggestions regarding model training, particularly the concept of a **continuous reward function**. This idea was highly inspiring, and we conducted corresponding experiments to investigate whether it could yield performance gains. We answer your questions below.
>
> ## Why single-step RL?
>
> The Stockfish analysis, which we use as the oracle, searches through subsequent multiple moves (we set the depth to 20). This implies that the optimal move for the current board state (derived from Stockfish analysis) has already considered many future steps; it is not limited to the immediate next move. While multi-step RL methods, such as self-play, might align more intuitively with the nature of Chess RL, resource limitations prevented us from conducting long-context post-training, thus leading us to primarily adopt single-step RL for training.
>
> ## Blindfold Play Mode
>
> Our designed Blindfold Play mode does not provide the model with direct information about the current board state. Similar to how a human player reconstructs the board mentally from move history, the model must infer the board from the complete move history provided via multi-turn dialogue. For instance, in the first turns: "You played e2e4, opponent played e7e5..." The model is required to: (1) **Reconstruct the board** from the dialogue history (which demands high multi-turn dialogue memory capacity) and (2) **Infer the optimal move** (which requires reasoning). This mode places a high demand on the model's overall memory and reasoning abilities.
>
> ## Can the reward function be continuous?
>
> We experimented with a continuous reward function. Building upon the same SFT model, we trained it using **10k chess RL samples** (due to the high CPU cost of Stockfish analysis required for continuous reward, we initially used 10k data points). Our proposed continuous reward is:
>
> $Reward = 1 - \frac{Rank_{move}}{len(legal\_{moves})} + 0.1 \times reward_{format}$
>
> In simple terms, the worst-ranked legal move receives a reward of 0, and the top-ranked move receives a reward of 1. Additionally, a $0.1$ format reward is added for valid moves. The performance of the model on the single-step move selection task is shown below:
>
> | | LR (%) | TR (%) | MAR (%) |
> | :--- | :---: | :---: | :---: |
> | With Legal Moves | 90.3 | 12.7 | -9.8 |
> | Without Legal Moves | 84.0 | 10.8 | -14.7 |
>
> > **Table 1: Performance of Qwen3-8B-Chess-Continuous-Reward**
> >
> > *(Note: The model trained with the continuous reward function performs significantly worse than the model trained with the discrete reward.)*
>
> The results indicate that the model trained with the continuous reward performs poorly, significantly trailing the discrete reward model. We hypothesize that in the game of chess, only learning the few **best moves** is effective. Making the model learn the relative ranking between all legal moves might be unnecessary and could even lead to reward hacking (where the model might not fully commit to exploring the very best moves).

---

> ### Author Response · Authors · 2025-11-26
> **Part 2**
>
> ## Generalization Experiments
>
> We used the math and code datasets from the following source: [https://huggingface.co/datasets/PRIME-RL/Eurus-2-RL-Data](https://huggingface.co/datasets/PRIME-RL/Eurus-2-RL-Data). We conducted experiments with five different RL configurations based on the Chess-SFT-Stage2 model:
> 1. 10k Math-RL
> 2. 10k Code-RL
> 3. 5k Math + 5k Code-RL
> 4. 8k Math + 2k Chess-RL
> 5. 3.3k Math + 3.3k Code + 3.3k Chess-RL
>
> We also conducted RL experiments with 10k Math samples on the original Qwen3-8B model.
>
> The primary external benchmarks we observed are **LiveCodeBench** (for code), **AIME2025** (for mathematics), and **ZebraLogic** (for logical reasoning), as shown in Table 2.
>
> | **Model Variant** | LiveCodeBench | AIME2025 | ZebraLogic |
> | :--- | :---: | :---: | :---: |
> | **Qwen3-8B (Baseline)** | 25.19 | 18.61 | 25.90 |
> | **I. Chess Training Only** | | | |
> | &nbsp;&nbsp; Qwen3-8B-Chess-SFT-Stage2 | 27.48 | 15.43 | 30.40 |
> | &nbsp;&nbsp; Qwen3-8B-Chess (SFT+RL) | 25.19 | 19.30 | 48.00 |
> | **II. Multi-Task RL (with Chess-SFT)** | | | |
> | &nbsp;&nbsp; +Math-RL | 25.19 | 19.70 | 42.40 |
> | &nbsp;&nbsp; +Code-RL | 32.82 | 16.21 | 32.30 |
> | &nbsp;&nbsp; +Math+Code-RL | 28.24 | 18.61 | 38.30 |
> | &nbsp;&nbsp; +Math+Chess-RL | 26.72 | 19.32 | 39.00 |
> | &nbsp;&nbsp; +Math+Code+Chess-RL | 30.53 | 17.96 | 38.00 |
> | **III. Multi-Task RL (without Chess-SFT)** | | | |
> | &nbsp;&nbsp; +Math-RL | 25.19 | 21.30 | 28.00 |
>
> > **Table 2: Performance of Qwen3-8B and its Post-Trained Variants on External Benchmarks**
>
> **Key Observations on Generalization:**
>
> 1.  Models fine-tuned via Chess Supervised Fine-Tuning (SFT), regardless of whether they subsequently undergo Code, Math, or Chess Reinforcement Learning (RL), consistently demonstrate a degree of generalization ability to the logical reasoning benchmark, ZebraLogic. Specifically, models that underwent Chess SFT show a significant improvement in their ZebraLogic scores after the RL phase.
> 2.  Incorporating a certain amount of Chess data into the RL dataset contributes to the generalization of Code capabilities, particularly on the LiveCodeBench. Comparisons (e.g., Math + Code + Chess - RL / Math + Chess - RL versus Math + Code - RL / Math - RL) reveal that RL models augmented with Chess data achieve an improvement in their LiveCodeBench scores.
>
> We have updated the performance on other benchmarks in the latest version of the PDF for your review.
>
> ## Zebralogic Case Study Explanation
>
> In the updated PDF, we have added two additional puzzles to supplement the case study in Appendix G.3. All puzzles illustrate that the model's reasoning *before* chess training was often superficial, frequently exhibiting steps skipped or known conditions forgotten. The model's reasoning *after* post-training in chess is not confined to a specific format but rigorously follows known conditions step-by-step, with demonstrable adjustments when errors occur (Puzzle 3). This indicates that the chess post-training enhances the model's reasoning process, making it more rigorous and leading to higher-quality answers.
>
> ## Performance Degradation from SFT-Stage2
>
> 1.  It is generally normal for a model fine-tuned on domain-specific data (SFT) to experience performance degradation in other, non-target domains. Since we exclusively trained on Chess, the observed performance drop on CruxEval (Code) is an expected outcome.
> 2.  We previously attempted to bypass SFT and proceed directly to RL, but the model was unable to roll out valid answers, resulting in an extremely low initial reward ( $< 0.05$). The RL training failed to converge, and the attempt was abandoned. However, whether direct RL could yield better reasoning performance warrants further investigation.
>
> Thank you once again for your meticulous review. Should you have any further questions, we will be glad to provide clarification.

---

### Official Review · Reviewer_4csM · 2025-11-02

**Soundness:** 2
**Presentation:** 3
**Contribution:** 2
**Rating:** 2
**Confidence:** 4

**Summary:**

The paper introduces ChessArena, a benchmark framework for evaluating the strategic reasoning capabilities of large language models (LLMs) using the game of chess. It proposes several play modes (Bullet, Blitz, Standard, and Blindfold) and fine-grained subtasks (basic rule understanding, move selection, and puzzle solving). The authors run extensive comparisons across over a dozen LLMs and also fine-tune Qwen3-8B to improve chess-related reasoning. Results suggest that current LLMs perform poorly in chess compared to even simple rule-based engines.

**Strengths:**

1.The paper provides a comprehensive benchmark, including several models and scoring metrics.
2.The experiments cover a broad range of LLMs under consistent conditions, providing useful insight for current model weaknesses in symbolic game reasoning.

**Weaknesses:**

1.The core idea that testing LLM through chess is not a novel concept, as there are already multiple earlier works like AlphaZero.
2.The author does not provide a clear and comprehensive analysis of why the model failed in chess.
3.Only fine-tuning on Qwen3-8B and showing the not-obvious gains.
4.Statements like “ChessArena reveals the reasoning limits of LLMs” are overreaching given that the benchmark measures a very narrow skill.

**Questions:**

1.Could the author comprehensively state the difference between your method and the current other game arena and general reasoning benchmark?
2.What qualitative reasoning patterns emerge from “thinking-enabled” models (O3, Gemini-Pro)?

---

> ### Author Response · Authors · 2025-11-26
> **Response to Reviewer 4csM**
>
> We sincerely appreciate the insightful and valuable feedback and questions raised by the reviewer. We address each point below.
>
> # Weaknesses
>
> ## Weakness 1
> ```
> The core idea that testing LLM through chess is not a novel concept, as there are already multiple earlier works like AlphaZero.
> ```
>
> We agree that chess has long been considered a solved problem since DeepBlue[1], but the focus of our study is fundamentally different. Our goal is **not** to achieve a State-of-the-Art (SOTA) chess model (which is unrealistic through LLMs now). Instead, we leverage chess as a challenging domain to evaluate the **strategic reasoning**, **instruction-following**, and **multi-turn dialogue memory** capabilities of Large Language Models (LLMs). These abilities are crucial for the advancement of modern LLMs. **ChessArena** establishes a deep and challenging chess benchmark for assessing these capabilities. Furthermore, we contribute a dedicated post-training dataset and code, hoping to contribute to the future development of LLMs.
>
> ## Weakness 2
> ```
> The author does not provide a clear and comprehensive analysis of why the model failed in chess.
> ```
>
> Thank you for raising this very important point. In the original manuscript, we did briefly discuss reasons for model failure in chess, such as insufficient instruction-following (failing to output in the required format) and limited reasoning ability (failing to infer the optimal move). We have updated our PDF to include a **more detailed and comprehensive analysis** of model failure in the chess domain. Please refer to **Appendix G.1** for the specific details.
>
> ## Weakness 3
> ```
> Only fine-tuning on Qwen3-8B and showing the not-obvious gains.
> ```
>
> Please refer to the **General Response**.
>
> ## Weakness 4
> ```
> Statements like “ChessArena reveals the reasoning limits of LLMs” are overreaching given that the benchmark measures a very narrow skill.
> ```
>
> We acknowledge that ChessArena is specifically a benchmark for evaluating models' **strategic reasoning in the context of chess**. Model reasoning ability is indeed a broad concept that includes many other facets, such as mathematical reasoning (e.g., [2]), code reasoning (e.g., [3]), and abstract strategic reasoning (e.g., ZebraLogic [4]). These existing benchmarks have made significant contributions to the community. However, **chess strategic reasoning** remains a vital component in assessing the overall intelligence of LLMs. We believe our statement accurately reflects the limits revealed within this important, complex, and strategic domain.
>
> References:
>
> [1] Campbell, Murray, A. Joseph Hoane Jr, and Feng-hsiung Hsu. "Deep blue." Artificial intelligence 134.1-2 (2002): 57-83.
>
> [2] MAA. American invitational mathematics examination - aime 2025, 2025. URL https://maa.org/math-competitions/american-invitational-mathematics-examination-aime.
>
> [3] Naman Jain, King Han, Alex Gu, Wen-Ding Li, Fanjia Yan, Tianjun Zhang, Sida Wang, Armando Solar-Lezama, Koushik Sen, and Ion Stoica. Livecodebench: Holistic and contamination free evaluation of large language models for code. arXiv preprint arXiv:2403.07974, 2024.
>
> [4] Bill Yuchen Lin, Ronan Le Bras, Kyle Richardson, Ashish Sabharwal, Radha Poovendran, Peter Clark, and Yejin Choi. Zebralogic: On the scaling limits of llms for logical reasoning. arXiv preprint arXiv:2502.01100, 2025.

---

> > ### Author Response · Authors · 2025-11-26
> > **Part 2**
> >
> > ## Question 1
> > ```
> > Could the author comprehensively state the difference between your method and the current other game arena and general reasoning benchmark?
> > ```
> >
> > We distinguish our work from existing benchmarks as follows:
> >
> > * **Game Arenas:** The work [5] is a concurrent work that initially used a head-to-head chess competition evaluation approach. However, the play mode of this work is limited to blitz.
> >
> > * **General Reasoning Benchmarks:** Previous commonly used reasoning benchmarks (e.g., [4], [6]) are often limited to a **Question/Answer (Q/A)** format. This format is insufficient for evaluating a model's performance under increasing pressure and across multiple dialogue turns. They also face potential data contamination issues.
> >
> > In contrast, our **ChessArena** serves as an entry point for assessing LLM reasoning ability via chess and offers several distinct advantages:
> >
> > 1.  **Diverse Evaluation Modes:** We include multiple play modes to test models under **stress scenarios** (e.g., *Bullet* mode, which disallows reasoning tokens) and with **increasing dialogue turns** (*Blindfold* mode, which necessitates strong multi-turn memory).
> > 2.  **Dataset Contribution:** We contribute a high-quality **Chess Reasoning Training Dataset** specifically designed for improving strategic reasoning in LLMs. According to the experiments in **Appendix G.3** of the updated PDF, our results indicate that chess SFT data contributes to some extent in enhancing the model's strategic reasoning capabilities.
> >
> > ## Question 2
> > ```
> > What qualitative reasoning patterns emerge from “thinking-enabled” models (O3, Gemini-Pro)?
> > ```
> >
> > We used the official APIs for O3 and Gemini-Pro, which unfortunately **do not provide access to the internal reasoning processes**. Consequently, it is challenging to perform a qualitative analysis of why O3 and Gemini-Pro exhibit superior performance. However, the official API for **Doubao-seed-1-6-thinking** *does* allow access to its internal reasoning. Our analysis reveals that Doubao-seed-1-6-thinking invests significantly more tokens in the inference phase (averaging >5,000 tokens vs. ~500 for non-thinking models like GPT-4.1 and DeepSeek-V3). These additional tokens are not merely verbose; they facilitate a rigorous generate-then-verify loop. The model explicitly simulates potential move sequences (look-ahead) to check for latent threats and validates the robustness of its current strategy. This iterative self-correction mechanism allows the model to filter out suboptimal moves that non-thinking models might select due to shallow reasoning depths.
> >
> > References:
> > [5] Andrew Lee, Bo Chang Antonio Gulli, Bovard Doerschuk-Tiberi Bob Fraser, Chris Prichard Chad Woodford, Bob Fraser, et al. Chess text input. https://www.kaggle.com/benchmarks/kaggle/chess-text, 2025. Google DeepMind, Google Cloud, Kaggle.
> >
> > [6] Dheeru Dua, Yizhong Wang, Pradeep Dasigi, Gabriel Stanovsky, Sameer Singh, and Matt Gardner. Drop: A reading comprehension benchmark requiring discrete reasoning over paragraphs. In Proceedings of the 2019 Conference of the North American Chapter of the Association for Computational Linguistics: Human Language Technologies, Volume 1 (Long and Short Papers), pp. 2368–2378, 2019.

---

### Official Review · Reviewer_TdZ7 · 2025-11-02

**Soundness:** 4
**Presentation:** 3
**Contribution:** 3
**Rating:** 4
**Confidence:** 4

**Summary:**

ChessArena is a unified testbed for measuring LLMs’ strategic reasoning through full chess games, with a built-in Glicko ranking system and four play modes (Bullet, Blitz, Standard, Blindfold) to probe speeded decision-making, long-horizon memory, and rule following. The platform also includes fine-grained diagnostics (basic board understanding, single-move quality vs. engine, and multi-step puzzles). In a large evaluation of 13+ LLMs over 800+ games, no model beats the amateur-level engine Maia-1100, and some even fail to outperform a random legal-move baseline, revealing persistent weaknesses in legality/format compliance, tactical selection, and long-range consistency.

**Strengths:**

1. Four complementary play modes (Bullet/Blitz/Standard/Blindfold) isolate different capabilities (speeded decision-making, reasoning allowance, and memory-only board reconstruction in Blindfold), which is an original capability-oriented decomposition.
2. Clear, interoperable interfaces (FEN for state; UCI/SAN for moves), scalable competition pipeline, and engine tooling (Stockfish for analysis; Maia-1100 and a Random baseline as anchors) make the setup robust and reproducible.

**Weaknesses:**

1. The arena evaluates only chess, so it is unclear how results transfer to broader strategic or multi-modal settings.
2. The paper contrasts models with human/engine anchors, but the human participant N, expertise, and variance are not clearly reported in the visible sections, limiting interpretation of the “gap.”
3. Post-training is validated on only a very small set of base models (Qwen3-8B only), making it hard to support a claim of general effectiveness. The observed gains may stem from family-specific characteristics or scale effects, rather than representing a stable benefit of the training paradigm itself.

**Questions:**

1. In Bullet/Blitz/Standard/Blindfold, how much of the gap is due to token budget or output-format differences? Please add a format-invariant comparison (e.g., fixed JSON “move” field with CoT stored separately) and report legality/format error rates as nuisance variables.
2. Will you release pairing logs, instance IDs, and trace-to-metric scripts (legality, top-move, rating computation), plus a minimal Docker? A short leaderboard policy (submission format, anti-overfit checks) would encourage adoption.

---

> ### Author Response · Authors · 2025-11-26
> **Response to Reviewer TdZ7**
>
> We sincerely thank the reviewer for your meticulous reading and constructive feedback on our manuscript. We have carefully considered all the suggestions and questions, and provide our detailed responses below.
>
> ---
> ```
> Weakness 1: The arena evaluates only chess, so it is unclear how results transfer to broader strategic or multi-modal settings.
> ```
> We initiated the construction of **ChessArena** with the explicit goal of establishing a deeper and more focused evaluation benchmark specifically for the game of Chess. This focus led us to devise various distinct play modes and to decompose the models' chess strategic reasoning capability through a fine-grained evaluation methodology.
>
> We argue that Chess, as a domain, is sufficiently rich to evaluate a model's **strategic reasoning** thoroughly, which is evidenced by the comprehensive evaluation and analysis we conducted. While benchmarks like GT-Bench[1] or GameArena[2] aim for broader coverage across multiple games, our contribution lies in the **depth of evaluation for a single game** and analysis achieved within the specialized context of Chess.
>
> ```
> Weakness 2: the paper contrasts models with human/engine anchors, but the human participant N, expertise, and variance are not clearly reported in the visible sections, limiting interpretation of the “gap.”
> ```
>
> In **ChessArena**, our human-level baseline is Maia-1100, a human-like chess engine that predicts the move an average human would make in a given position. Introduced in Literature [3], Maia-1100 builds on AlphaZero's core architecture but is uniquely trained to emulate human move selection rather than achieve superhuman play.
>
> On the official Lichess platform, Maia-1100's real Blitz Elo rating is approximately $1500$ (URL: https://lichess.org/@/maia1/perf/blitz), though this rating varies over time. This score roughly corresponds to the average human player's skill level. Due to resource constraints, **no human subjects** participated in the games within the **ChessArena** evaluation itself.
>
> ```
> Weakness 3: Post-training is validated on only a very small set of base models (Qwen3-8B only), making it hard to support a claim of general effectiveness. The observed gains may stem from family-specific characteristics or scale effects, rather than representing a stable benefit of the training paradigm itself.
> ```
>
> See **General Response** please.
>
> References:
>
> [1] Andrew Lee, Bo Chang Antonio Gulli, Bovard Doerschuk-Tiberi Bob Fraser, Chris Prichard Chad Woodford, Bob Fraser, et al. Chess text input. https://www.kaggle.com/ benchmarks/kaggle/chess-text, 2025. Google DeepMind, Google Cloud, Kaggle.
>
> [2] Jinhao Duan, Renming Zhang, James Diffenderfer, Bhavya Kailkhura, Lichao Sun, Elias Stengel- Eskin, Mohit Bansal, Tianlong Chen, and Kaidi Xu. Gtbench: Uncovering the strategic reasoning capabilities of llms via game-theoretic evaluations. Advances in Neural Information Processing Systems, 37:28219–28253, 2024.
>
> [3] Reid McIlroy-Young, Siddhartha Sen, Jon Kleinberg, and Ashton Anderson. Aligning superhuman ai with human behavior: Chess as a model system. In Proceedings of the 26th ACM SIGKDD international conference on knowledge discovery \& data mining, pp. 1677–1687, 2020.

---

> > ### Author Response · Authors · 2025-11-26
> > **Part 2**
> >
> > ```
> > Questions 1: In Bullet/Blitz/Standard/Blindfold, how much of the gap is due to token budget or output-format differences? Please add a format-invariant comparison (e.g., fixed JSON“move”field with CoT stored separately) and report legality/format error rates as nuisance variables.
> > ```
> >
> > This is a highly pertinent question. We address the potential influence of token budget and output format differences as follows:
> >
> > **Token Budget:**
> >
> > In our ChessArena, we adopt a highly permissive configuration for the maximum token limit to ensure that models are not inhibited from playing due to constraints on reasoning tokens. Specifically, the max tokens are set to 4096 for non-thinking models and 16384 for Thinking-enabled models. Furthermore, the inclusion of **multiple retries** for output generation substantially increases the effective token budget for the model's response.
> >
> > **Output-Format Differences:**
> > We accommodate model outputs in both **UCI and SAN** formats, which are the two most common official Chess move notations. The required move output is enclosed within a code block (i.e., \`\`\` e2e4 \`\`\`). Similarly, the use of **multiple retry attempts** is designed to mitigate the influence of minor output format errors. Should a model fail to produce a legal move after several retries, we deem this to be a fundamental issue of instruction-following inherent to the model, which is included within the scope of our evaluation.
> >
> > **Report Legality/Format Error Rates:**
> > We have updated the PDF to include a dedicated section reporting these nuisance variables. Tables 11, 12, 13, and 14 in the revised PDF now detail the models' legality and format error rates, alongside the various termination conditions (e.g., checkmate or forfeit). A comprehensive analysis of these results is provided in **Appendix F.1**.
> >
> > ```
> > Questions 2: Will you release pairing logs, instance IDs, and trace-to-metric scripts (legality, top-move, rating computation), plus a minimal Docker? A short leaderboard policy (submission for-mat, anti-overfit checks) would encourage adoption.
> > ```
> >
> > Yes! We have packaged all the complete match history, instance IDs, and the corresponding scripts used for metric computation (including rating calculation) into a minimal **Docker image**.
> >
> > **Accessing the Docker Image**
> >
> > The Docker image, packaged as a $\text{tar.gz}$ archive, is available for download via the provided anonymous URL. To access and run the environment, please follow the steps below:
> >
> > 1.  **Download the Archive:**
> >     The $\text{tar.gz}$ archive of the Docker image can be obtained using the following link:
> >     (URL: https://drive.google.com/file/d/1XAcoc4_Ij65cK6ICBgBZv7nwNbNes6fp/view?usp=drive_link)
> >
> > 2.  **Decompress the Archive:**
> >     Decompress the downloaded $\text{tar.gz}$ file to obtain the `.tar` image file:
> >     ```bash
> >     gunzip chessarena_for_rebuttal.tar.gz
> >     ```
> >
> > 3.  **Import the Docker Image:**
> >     Load the Docker image into your local Docker environment using the resulting `.tar` file:
> >     ```bash
> >     docker load -i chessarena_for_rebuttal.tar
> >     ```
> >
> > 4.  **Run the Container:**
> >     Execute the Docker image. Upon startup, the container will automatically initiate the ELO rating calculation and display the results:
> >     ```bash
> >     docker run -it --rm chessarena:v0.1
> >     ```
> >
> > Alternatively, you may enter the container to examine the files (including all match history log files) using:
> >
> > ```bash
> > docker run -it --rm chessarena /bin/bash
> > ```
> >
> > All simulation record logs are specifically located in the
> > ```
> > ./simulation_record/
> > ```
> > folder within the container.
> >
> > We will also draft a short leaderboard policy (including submission format and guidelines for anti-overfitting checks) to facilitate community adoption and fair competition.
> >
> > ---
> >
> > Once again, thank you very much for your thoughtful review and invaluable suggestions. Please feel free to raise any further questions; we are committed to addressing them thoroughly.

---

### Author Response · Authors · 2025-11-26
**General Response**

Both Reviewer TdZ7 and Reviewer 4csM highlighted the limitation of our study, noting that the post-training was restricted to a single model family. We acknowledge that the initial validation of our post-training approach was limited to the Qwen3-8B model family. To address this concern regarding generality, we have since conducted an additional post-training experiment on **Seed-Coder-8B-Instruct** (URL: https://huggingface.co/ByteDance-Seed/Seed-Coder-8B-Instruct) (a model belonging to the Llama model family). The specific results are as follows:

| Model | Elo |
| --- | --- |
| Qwen3-8B | 1335 |
| Qwen3-8B-Chess(Post-trained) | 1776 (+441) |
| Seed-Coder-8B-Instruct | 1009 |
| Seed-Coder-8B-Chess(Post-trained) | 1614(+605) |

These extended results will be incorporated into the revised manuscript to better support the general effectiveness of our training paradigm.

---

### Author Response · Authors · 2025-12-02
**Rebuttal Summary**

Dear Area Chairs and Program Chairs,

We provide a summary of our rebuttal updates and additional experiments below:

**1. General Response (Reviewers TdZ7 & 4csM): Model Family Diversity**

Addressing the common concern raised by Reviewers TdZ7 and 4csM regarding the limitation of post-training on a single model family, we have conducted additional post-training experiments using Seed-Coder-8B-Instruct (Llama family) in our General Response. The results demonstrate a significant improvement in capabilities, validating the method across different model families.

**2. Response to Reviewer TdZ7**
* **Evaluation Sub-metrics:** Regarding the request for specific sub-metrics (e.g., Legality/Format Error Rates), we have updated the paper to include these details in **Appendix F.1**.
* **Reproducibility (Docker Image):** As suggested, we have packaged the ChessArena environment into a Docker image. We provided the image address and specific execution commands in our response. The running logs for ChessArena competitions are also accessible within the image for verification.

**3. Response to Reviewer 4csM**
* **Deep Analysis of LLM Performance:** To address the concern regarding the lack of in-depth analysis, we have added a comprehensive analysis of the LLM's performance in the revised **Appendix G.1**.

**4. Response to Reviewer UYks**
* **Continuous vs. Sparse Rewards:** We conducted experiments comparing continuous rewards against our baseline. We found that continuous rewards yield inferior performance. We hypothesize that in Chess, only a few optimal moves are critical for controlling the game state; therefore, learning the relative ranking between sub-optimal moves is less effective than directly learning the optimal moves.
* **Generalization Experiments:** We performed additional experiments detailed in **Appendix G.3**:
    * We found that generalization on ZebraLogic primarily depends on Chess SFT, validating the effectiveness of this stage.
    * We observed that incorporating Chess data into Code and MATH RL datasets improves the model's coding capabilities(i.e., LiveCodeBench), suggesting the potential benefit of mixing Chess RL data during training.
* **ZebraLogic Case Study:** We added two puzzle cases in **Appendix G.3**. These examples demonstrate that the model with Chess post-training exhibits more rigorous reasoning, avoiding skipped steps or ignored constraints, and displays self-verification behaviors. Consequently, the model's ability to solve Large and X-Large puzzles is significantly enhanced.

---

### Note · Authors · 2026-01-06

I have read and agree with the venue's withdrawal policy on behalf of myself and my co-authors.